# LEVERAGING COORDINATE MOMENTUM IN SIGNSGD AND MUON: MEMORY-OPTIMIZED ZERO-ORDER LLM FINE-TUNING

## ABSTRACT

Fine-tuning Large Language Models (LLMs) is essential for adapting pre-trained models to downstream tasks. Yet traditional first-order optimizers such as Stochastic Gradient Descent (SGD) and Adam incur prohibitive memory and computational costs that scale poorly with model size. In this paper, we investigate zero-order (ZO) optimization methods as a memory- and compute-efficient alternative, particularly in the context of parameter-efficient fine-tuning techniques like LoRA. We propose `JAGUAR SignSGD`, a ZO momentum-based algorithm that extends ZO SignSGD, requiring the same number of parameters as the standard ZO SGD and only $\mathcal{O}(1)$ function evaluations per iteration. To the best of our knowledge, this is the first study to establish rigorous convergence guarantees for SignSGD in the stochastic ZO case. We further propose `JAGUAR Muon`, a novel ZO extension of the Muon optimizer that leverages the matrix structure of model parameters, and we provide its convergence rate under arbitrary stochastic noise. Through extensive experiments on challenging LLM fine-tuning benchmarks, we demonstrate that the proposed algorithms meet or exceed the convergence quality of standard first-order methods, achieving significant memory reduction. Our theoretical and empirical results establish new ZO optimization methods as a practical and theoretically grounded approach for resource-constrained LLM adaptation. Our code is available at https://anonymous.4open.science/r/zo_jaguar

## 1 INTRODUCTION

Fine-tuning pre-trained Large Language Models (LLMs) has become the standard technique in modern natural language processing (Howard & Ruder, 2018; Zhang et al., 2019; 2024a; Lester et al., 2021), enabling rapid adaptation to diverse downstream tasks with minimal labelled data (Raffel et al., 2020; Sanh et al., 2021; Zaken et al., 2021). These models, often trained on massive corpora, achieve state-of-the-art results when fine-tuned on specific applications, including question answering, summarization, and dialogue generation. The fine-tuning setup can be considered as a stochastic unconstrained optimization problem of the form

$$f^* := \min_{x \in \mathbb{R}^d} \{ f(x) := \mathbb{E}_{\xi \sim \mathcal{D}} [f(x, \xi)] \}, \tag{1}$$

where $x$ are parameters of the fine-tuned LLM, $\mathcal{D}$ is the data distribution available for training, and $f(x, \xi)$ is the loss on data point $\xi$.

The de facto standard for solving (1) is the use of First-Order (FO) optimization methods. These approaches assume access to the stochastic gradient $\nabla f(x, \xi)$. Classical FO methods, such as Stochastic Gradient Descent (SGD) (Amari, 1993) and Adam (Kingma & Ba, 2014), remain the most widely used techniques for model adaptation due to their efficiency and compatibility with the backpropagation algorithm. Nevertheless, in contemporary fine-tuning tasks, alternative FO algorithms are often preferred.

A recent trend in optimization for LLMs is to represent optimization parameters in matrix form rather than as vectors (Bernstein & Newhouse, 2024b;a; Pethick et al., 2025). Algorithms such as Shampoo (Gupta et al., 2018) and SOAP (Vyas et al., 2024) have demonstrated superior performance on LLM training tasks compared to Adam and SGD (Dahl et al., 2023), which operate in an element-wise

manner and do not utilize the underlying structure of the model parameters. Currently, the canonical matrix-based optimization algorithm is Muon (Jordan et al., 2024; Liu et al., 2025; Li & Hong, 2025), which integrates the principles of Shampoo and SOAP but does not employ any preconditioning matrices (Jordan et al., 2024). The central idea of this method is to project the gradient at each iteration onto the space of semi-orthogonal matrices using the Newton–Schultz algorithm (Bernstein & Newhouse, 2024b).

However, as LLMs continue to scale, the backpropagation procedure, necessary for FO methods, becomes increasingly expensive in terms of memory consumption. For instance, the memory cost of computing gradients during the training of OPT-13B is reported to be more than an order of magnitude larger than that of inference (Zhu et al., 2023b). This imbalance poses a serious bottleneck for deploying LLM fine-tuning in resource-constrained environments such as edge devices (Zhu et al., 2023a; Gao et al., 2024), consumer-grade GPUs (Liao et al., 2024; Yin et al., 2023), or large-scale distributed settings (Han et al., 2015). To overcome these limitations, researchers are exploring various approaches to reduce the size of the required optimizer statistics, especially in comparison to adaptive methods such as Adam. One such approach is the SignSGD algorithm, initially developed for distributed optimization (Yang et al., 2020), but which has also proven effective in LLM fine-tuning (Peng et al., 2025), owing to its simplicity, memory efficiency, and surprising empirical effectiveness across a range of adaptation tasks (Jin et al., 2020; Mengoli et al., 2025). SignSGD was first rigorously analyzed in the FO setting by (Bernstein et al., 2018) and (Balles & Hennig, 2017). Minimal memory usage and straightforward hyperparameter tuning make SignSGD an attractive choice for memory-constrained fine-tuning of LLMs ($\sim 4/3\times$ memory usage compared to Adam). Beyond SignSGD, other FO methods also target memory reduction. AdaFactor (Shazeer & Stern, 2018) was among the first, lowering memory usage by storing a single value per block ($\sim 4/3\times$). Additional techniques include quantizing optimizer states to lower-precision formats (Dettmers et al., 2021; Li et al., 2023) ($\sim 4/3\times$ and $\sim 16/9\times$ respectively), fusing the backward pass with optimizer updates (Lv et al., 2023) ($\sim 4/3\times$) and low-rank optimizer-states decompositions such as GaLore (Zhao et al., 2024) (up to $\sim 3/2\times$), further decreasing memory demands during training.

Nevertheless, the most memory-efficient methods are based on the Zero-Order (ZO) optimization technique, which avoids backpropagation entirely by estimating gradients using only forward passes. This flexibility allows us to treat the model as a black box, optimizing performance with minimal assumptions about its architecture or implementation details. Recent studies (Malladi et al., 2023) have demonstrated the practical benefits of this approach: for example, the MeZO algorithm applies classical ZO SGD (Ghadimi & Lan, 2013) to fine-tune LLMs while maintaining four times lower memory requirements than traditional FO methods (Malladi et al., 2023) ($\sim 10\times$ compared to Adam (Zhang et al., 2024b)). In ZO methods it is assumed that we only have access to the values of the stochastic function $f(x, \xi)$ from (1) (Flaxman et al., 2005; Ghadimi & Lan, 2013). Within LLMs pretraining or fine-tuning context, oracles are forward passes with small perturbations in parameters of the model. To estimate gradients, authors use finite differences:

$$\nabla f(x, \xi) \approx \frac{f(x + \tau e, \xi) - f(x - \tau e, \xi)}{2\tau} e, \tag{2}$$

where $\tau > 0$ is a small number, frequently referred to as a smoothing parameter, and $e \in \mathbb{R}^d$ is some random vector (Nesterov & Spokoiny, 2017; Duchi et al., 2015; Malladi et al., 2023; Zhang et al., 2024b). In the next section, we provide review about different ZO optimization methods, that somehow utilize formula (2).

## 2 RELATED WORK AND OUR CONTRIBUTIONS

**ZO gradient estimators.** The simplest zero-order gradient estimator employs the estimate (2) as the stochastic gradient. However, even this approach presents specific challenges, particularly regarding the selection of an appropriate distribution from which to sample the random vector $e$. The most commonly employed distributions include a uniform sampling over the unit sphere: $e \sim RS(1)_{\|\cdot\|}^d$ (Flaxman et al., 2005; Nesterov & Spokoiny, 2017), a Gaussian distribution with zero mean and identity covariance matrix: $e \sim \mathcal{N}(0, I)$ (Nesterov & Spokoiny, 2017; Ghadimi & Lan, 2013), and standard basis one-hot vectors (Duchi et al., 2015; Shamir, 2013). Also, some papers (Lian et al., 2016; Sahu et al., 2019; Akhtar & Rajawat, 2022) utilize the so-called full coordinate estimate, which approximates the gradient across all basis vectors. However, this approach requires $\mathcal{O}(d)$ calls to the

zero-order oracle, making it impractical for large-scale fine-tuning tasks. Despite the prevalence of these approaches, alternative and more complicated sampling strategies have also been explored.

In (Roberts & Royer, 2023; Nozawa et al., 2025), the authors explore low-dimensional perturbations within random subspaces. The central concept of random subspace methods involves generating the perturbation vector $e$ within a subspace spanned by a projection matrix $P \in \mathbb{R}^{d \times r}$ and a low-dimensional random vector $\tilde{e} \in \mathbb{R}^r$: $e = P\tilde{e}$. Typically, $P$ and $\tilde{e}$ are sampled from a Gaussian distribution and $r \ll d$. The primary motivation for this method lies in the fact that gradients during the fine-tuning process exhibit a low-dimensional structure (Nozawa et al., 2025). In (Liu et al., 2024; Wang et al., 2024), the authors employ a masked random vector $e$, wherein at each iteration a random mask with $r$ non-zero elements $m_r \in \{0, 1\}^d$ is generated and applied element-wise to a Gaussian vector $e$. This procedure accelerates the optimization step, as only the parameters corresponding to the active entries in $m_r$ are updated, rather than the entire parameter set. In contrast, the authors of (Guo et al., 2024b) depart from random mask sampling at each iteration and instead select an optimal mask $m_r$ prior to training, according to a specific criterion. Consequently, the update rule (2) modifies only the parameters selected by the optimal mask during optimization.

In our approach, we do not utilize all coordinates of the random vector $e$ in each estimation of (2), instead, we select a single coordinate at each step similar to (Liu et al., 2024; Wang et al., 2024; Guo et al., 2024b). However, unlike previous works, we do not discard information from the remaining coordinates, but accumulate information from previous iterations. We employ the JAGUAR zero-order gradient estimation technique (Veprikov et al., 2024; Nazykov et al., 2024), which integrates the concept of sampling one-hot basis vectors with the utilization of a SAGA-like momentum update (Defazio et al., 2014). This approach facilitates convergence in the stochastic setting by leveraging memory from past iterations, while using the same amount of memory as standard zero-order methods like ZO SGD (MeZO) (Malladi et al., 2023). In the original paper (Veprikov et al., 2024), the authors do not incorporate a momentum parameter, discarding coordinate information from previous iterations. In contrast, we introduce a momentum parameter, $0 \leq \beta \leq 1$ (see Algorithms 1 and 2), which controls the utilization of gradients from past iterations. We demonstrate that adding this momentum $\beta$ allows the method to converge in the stochastic non-convex case (see Theorems 3.5 and 3.7).

**Momentum techniques.** Numerous zero-order methods in the literature incorporate momentum techniques in various forms. However, these approaches typically introduce multiple additional variables of dimension $d$. Since zero-order methods are often chosen for fine-tuning tasks to save memory, the inclusion of such extra variables becomes a critical limitation in these settings. In (Huang et al., 2022), authors use variance reduction technique SPIDER (Fang et al., 2018), that uses approximately $5d$ parameters: $2d$ for ZO gradients, $2d$ for model parameters and $1d$ for momentum. In (Chen et al., 2019; Jiang et al., 2024), the authors employ the Adam optimization technique (Kingma & Ba, 2014), which is frequently used for stochastic non-convex optimization problems (Chen et al., 2019; et al., 2024). However, this technique incurs a significant memory overhead, requiring $4d$ parameters. The paper (Reddy & Vidyasagar, 2023) utilizes classical heavy-ball momentum within a zero-order framework, provided, only demonstrating almost sure convergence to a constant in the non-convex setting. In our work, we successfully incorporated a momentum technique using only $2d + 1$ parameters and proved the convergence rate within the standard stochastic non-convex setting (see Algorithm 1 and Theorem 3.5). It is worth noting that numerous other zero-order techniques exist in the literature to achieve convergence when the function $f$ is convex (Gorbunov et al., 2022; Nesterov & Spokoiny, 2017; Duchi et al., 2015), satisfies conditions like PL (Reddy & Vidyasagar, 2023) or ABG (Rando et al., 2024), or in deterministic settings (Bergou et al., 2020). Since our focus is on fine-tuning problems, which fall under the stochastic non-convex case, we will not discuss these methods in detail.

**Matrix ZO optimization.** In the context of zero-order optimization, transitioning to matrix-valued parameters necessitates replacing the random vector $e \in \mathbb{R}^d$ in zero-order gradient approximation (2) with a random matrix $E \in \mathbb{R}^{m \times n}$, and correspondingly, projecting this matrix $E$ onto a semi-orthogonal space, as is done in the Muon algorithm (Jordan et al., 2024). Since the random matrix $E$ is typically drawn from a known distribution, it is possible to directly sample orthogonal matrices when computing the gradient estimator (2). A similar approach has previously appeared in the zero-order optimization literature (Chen et al., 2024); however, that work did not consider the Muon algorithm, but rather focused on sampling two Gaussian matrices $V \in \mathbb{R}^{m \times r}$ and $U \in \mathbb{R}^{n \times r}$ of rank $r \ll \min\{m, n\}$. This approach does not correspond to the decomposition of the random matrix $E$, as $E$ is almost surely of full rank. Additionally, alternative techniques for sampling low-rank

matrices have been proposed in the literature. For instance, in (Yu et al., 2024), a method analogous to the sampling of low-rank vectors described in (Roberts & Royer, 2023; Nozawa et al., 2025) is utilized. In our work, we extend our memory-efficient momentum method to the ZO version of the matrix-based Muon algorithm (Jordan et al., 2024) (see Algorithm 2 and Theorem 3.7), keeping the $2d + 1$ parameter efficiency while also broadening our analysis to more modern algorithms that leverage the matrix structure of parameters.

We present a summary of relevant results from the existing zero-order literature in Table 1.

Table 1: Summary of relevant results from the existing zero-order literature.

| | Method | Parameter Count | Convergence Rate Stochastic Non-convex Case | Momentum | Fine-tuning (LLM) Setup |
|---|---|---|---|---|---|
| **Vector Parameters** $\mathbf{x} \in \mathbb{R}^d$ | ZO-SGD (Ghadimi & Lan, 2013) | $\mathbf{2 \cdot d}$ | ✓ | ✗ | ✗ |
| | ZO-PSGD (Ghadimi et al., 2016) | $\mathbf{2 \cdot d}$ | ✓ | ✗ | ✗ |
| | ZO-SCD (Lian et al., 2016) [1] | $\mathbf{2 \cdot d}$ | ✓ | ✗ | ✗[2] |
| | ZO-SPIDER (Fang et al., 2018) | $\mathbf{5 \cdot d}$ | ✓ | ✓ | ✗ |
| | ZO-AdaMM (Chen et al., 2019) | $\mathbf{4 \cdot d}$ | ✓ | ✓ | ✗ |
| | ZO-SignSGD (Liu et al., 2019a) | $\mathbf{2 \cdot d}$ | ✗ ✓[3] | ✗ | ✗[4] |
| | Acc-ZOM (Huang et al., 2022) | $\mathbf{5 \cdot d}$ | ✓ | ✓ | ✗ |
| | DSFBSD (Roberts & Royer, 2023) | $(\mathbf{1 + r}) \cdot \mathbf{d}$ [5] | ✗ | ✗ | ✗ |
| | MeZO (Malladi et al., 2023) | $\mathbf{2 \cdot d}$ | ✗ | ✗ | ✓ |
| | ZO-ProxSTORM (Qian & Zhao, 2023) | $\mathbf{5 \cdot d}$ | ✓ | ✓ | ✗ |
| | HB ZO-SGD (Reddy & Vidyasagar, 2023) | $\mathbf{3 \cdot d}$ | ✗[6] | ✓ | ✗ |
| | Sparse ZO-SGD (Guo et al., 2024a) | $(\mathbf{2 + r}) \cdot \mathbf{d}$ [5] | ✗ | ✗ | ✓ |
| | Sparse MeZO (Liu et al., 2024) | $\mathbf{3 \cdot d}$ | ✗ | ✗ | ✓ |
| | LeZO (Wang et al., 2024) | $\mathbf{2 \cdot d}$ | ✗ | ✗ | ✓ |
| | ZO-AdaMU (Jiang et al., 2024) | $\mathbf{4 \cdot d}$ | ✓ | ✓ | ✓ |
| | ZO-SGD-Cons (Kim et al., 2025) | $\mathbf{2 \cdot d}$ | ✗ | ✗ | ✓ |
| | SGFM (Nozawa et al., 2025) | $(\mathbf{2 + r}) \cdot \mathbf{d}$ [5] | ✗ | ✗ | ✗ |
| | CompSGD (Kornilov et al., 2025) | $\mathbf{2 \cdot d}$ | ✗ ✓[3] | ✗ | ✓ |
| | **JAGUAR SignSGD** **Algorithm** 1 | $\mathbf{2 \cdot d + 1}$ | ✓ | ✓ | ✓ |
| **Matrix Parameters** $\mathbf{X} \in \mathbb{R}^{m \times n}$ | ZO-RMS (Maass et al., 2021) [7] | $\mathbf{2 \cdot mn}$ | ✗ ✓[3] | ✗ | ✗ |
| | MeZO (Malladi et al., 2023) | $\mathbf{2 \cdot mn}$ | ✗ | ✗ | ✓ |
| | LOZO (Chen et al., 2024) | $(\mathbf{m + n})\mathbf{r} + \mathbf{2 \cdot mn}$ [5] | ✓ | ✗ | ✓ |
| | SubZero (Yu et al., 2024) [8] | $(\mathbf{m + n + r})\mathbf{r} + \mathbf{2 \cdot mn}$ [5] | ✗ | ✗ | ✓ |
| | **JAGUAR Muon** **Algorithm** 2 | $\mathbf{2 \cdot mn + 1}$ | ✓ | ✓ | ✓ |

[1] Uses a full coordinate ZO estimator. [2] Considers asynchronous algorithms. [3] Convergence only to a neighborhood of the solution. [4] Addresses adversarial attacks in deep learning. [5] $r \ll d, m, n$ is a small number. [6] Only asymptotic convergence to a constant. [7] Assumes that parameters are symmetric matrices. [8] Assumes sparsity of parameters.

## 2.1 OUR CONTRIBUTIONS

While zero-order optimization methods have recently attracted attention for LLM fine-tuning, previous work has primarily focused on basic algorithms. In this paper, we broaden the scope of zero-order optimization by introducing advanced momentum techniques, specifically adapting the JAGUAR approach (Veprikov et al., 2024) to the SignSGD algorithm in the zero-order setting (see Algorithms 1). We consider this algorithm because SignSGD has demonstrated state-of-the-art performance in LLM fine-tuning tasks, outperforming even AdamW (Peng et al., 2025). Our key contributions are as follows:

- We provide the first convergence analysis in the stochastic non-convex setting for zero-order SignSGD with momentum (Algorithm 1 and Theorem 3.5), requiring only $2d + 1$ parameters and $\mathcal{O}(1)$ ZO oracle calls per iteration.

- We extend our memory-efficient momentum method to the Muon algorithm (Algorithm 2), introducing the first zero-order variant of Muon that preserves memory efficiency. We also establish its convergence rate in the stochastic non-convex setting (Theorem 3.7).

- We empirically evaluate the proposed zero-order methods on challenging LLM fine-tuning benchmarks, demonstrating their effectiveness and practical relevance.

## 3 MAIN RESULTS

### 3.1 PRELIMINARIES

**Notations.** We denote the $\ell_1$ and $\ell_2$ (Euclidean) norms of a vector $x \in \mathbb{R}^d$ as $\|x\|_1 := \sum_{i=1}^d |x_i|$ and $\|x\|_2^2 := \sum_{i=1}^d x_i^2$. Matrix-valued variables are denoted by capital letters. For matrices $X \in \mathbb{R}^{m \times n}$, we use the Schatten 1-norm ($\mathcal{S}_1$) and Schatten 2-norm ($\mathcal{S}_2$, Frobenius): $\|X\|_{\mathcal{S}_1} := \sum_{i=1}^d |(\Sigma_X)_{i,i}|$ and $\|X\|_{\mathcal{S}_2}^2 := \sum_{i=1}^d (\Sigma_X)_{i,i}^2 = \sum_{i=1}^m \sum_{j=1}^n X_{i,j}^2 =: \|X\|_F^2$ (the $\ell_1$ and $\ell_2$ norms of the eigenvalues of $X$), where $X = U_X \Sigma_X V_X^T$ is the reduced SVD of $X$. We define dot product between two vectors $x, y \in \mathbb{R}^d$ as $\langle x, y \rangle := x^T y$. For matrices $X, Y \in \mathbb{R}^{m \times n}$, we define $\langle X, Y \rangle := \text{tr}(X^T Y)$. We use the notation of the uniform distribution: $\text{Uniform}(\overline{1, d})$, where $\overline{1, d} := \{1, 2, \ldots, d\}$.

We now provide several assumptions that are necessary for the analysis.

**Assumption 3.1** (Smoothness)**.** The functions $f(x, \xi)$ are $L(\xi)$-smooth on the $\mathbb{R}^d$ with respect to the Euclidean norm $\|\cdot\|$, i.e., for all $x, y \in \mathbb{R}^d$ it holds that $\|\nabla f(x, \xi) - \nabla f(y, \xi)\|_2 \leq L(\xi) \|x - y\|_2$. We also assume that exists constant $L^2 := \mathbb{E}[L(\xi)^2]$.

**Assumption 3.2** (Bounded variance of the gradient)**.** The variance of the $\nabla f(x, \xi)$ is bounded with respect to the Euclidean norm, i.e., there exists $\sigma > 0$, such that for all $x \in \mathbb{R}^d$ it holds that $\mathbb{E}[\|\nabla f(x, \xi) - \nabla f(x)\|_2^2] \leq \sigma^2$.

We assume access only to a zero-order oracle, which returns a noisy evaluation of the function $f(x, \xi)$. Therefore, we are limited to using this noisy value $\hat{f}(x, \xi)$ in the estimation of the ZO gradient (2). This noise may originate not only from inherent randomness (stochastic noise), but also from systematic effects (deterministic noise), such as computer rounding errors. Therefore, we make a common assumption about the function $\hat{f}(x, \xi)$ returned by the oracle (Dvurechensky et al., 2021).

**Assumption 3.3** (Bounded oracle noise)**.** The noise in the oracle is bounded with respect to the Euclidean norm, i.e., there exists $\Delta > 0$, such that for all $x \in \mathbb{R}^d$ it holds that $\mathbb{E}\left[\left|\hat{f}(x, \xi) - f(x, \xi)\right|^2\right] \leq \Delta^2$.

Assumptions 3.1 and 3.2 are standard in the theoretical analysis of stochastic non-convex zero-order optimization problems (Guo et al., 2024b; Liu et al., 2024; Wang et al., 2024). Assumption 3.3 is also quite standard (Lobanov et al., 2023; Kornilov et al., 2023; Veprikov et al., 2024). However, Assumption 3.3 is sometimes omitted in the literature (Malladi et al., 2023; Zhang et al., 2024b), as it is commonly presumed that $\Delta = 0$, implying access to an ideal zero-order oracle. However, this assumption does not hold in practice, as numerical errors such as machine precision inevitably introduce a non-zero perturbation. Consequently, while $\Delta$ is typically small, it is never zero, which does not allow us to restore a true gradient along the direction $e$ in the estimation (2) if we set $\tau \to 0$.

### 3.2 ZERO-ORDER MOMENTUM SIGNSGD WITH JAGUAR GRADIENT APPROXIMATION

In this section, we introduce zero-order SignSGD algorithm with JAGUAR gradient approximation (Veprikov et al., 2024; Nazykov et al., 2024) and momentum of the form:

---

**Algorithm 1** Zero-Order Momentum SignSGD with JAGUAR (`JAGUAR SignSGD`)

---

1: **Parameters:** stepsize $\gamma$, momentum $\beta$, smoothing parameter $\tau$, number of iterations $T$.
2: **Initialization:** choose $x^0 \in \mathbb{R}^d$ and $m^{-1} = \mathbf{0} \in \mathbb{R}^d$.
3: **for** $t = 0, 1, 2, \ldots, T$ **do**
4:     Sample $i_t \sim \text{Uniform}(\overline{1, d})$
5:     Set one-hot vector $e^t$ with 1 in the $i_t$ coordinate
6:     Sample stochastic variable $\xi^t \sim \mathcal{D}$
7:     Set $\widetilde{\nabla}_{i_t} f(x^t, \xi^t) := \frac{\hat{f}(x^t + \tau e^t, \xi^t) - \hat{f}(x^t - \tau e^t, \xi^t)}{2\tau} \in \mathbb{R}$
8:
9:     Set $m_{i_t}^t = \beta m_{i_t}^{t-1} + (1 - \beta) \widetilde{\nabla}_{i_t} f(x^t, \xi^t)$ and $m_{i \neq i_t}^t = m_{i \neq i_t}^{t-1}$
10:     Set $x^{t+1} = x^t - \gamma \cdot \text{sign}(m^t)$
11: **end for**
12: **Return:** $x^{N(T)}$, where $N(T) \sim \text{Uniform}(\overline{1, T})$.

---

The gradient approximation employed in Algorithm 1 deviates from that of the original JAGUAR method, as we introduce a momentum variable $\beta$. The estimator from the original work can be recovered by setting $\beta = 0$.

We now present a lemma characterizing the closeness between the momentum variable $m^t$ from line 8 of Algorithm 1 and the true gradient $\nabla f(x^t)$.

**Lemma 3.4.** *Consider $m^t$ from line 8 of Algorithm 1. Under Assumptions 3.1, 3.2, 3.3 it holds that:*

$$\mathbb{E}\left[\left\|m^t - \nabla f(x^t)\right\|_2^2\right] = \mathcal{O}\left[\frac{d^3 L^2 \gamma^2}{(1-\beta)^2} + (1-\beta)d\sigma^2 + dL^2\tau^2 + \frac{2d\Delta^2}{\tau^2} + \left(1 - \frac{1-\beta}{d}\right)^t \left\|\nabla f(x^0)\right\|_2^2\right].$$

**Discussion.** This lemma closely parallels Lemma 1 from (Veprikov et al., 2024), with the key distinction that our analysis incorporates the momentum parameter $\beta$, which was not present in (Veprikov et al., 2024). The introduction of momentum is essential for proving convergence of algorithms such as SignSGD (Algorithm 1) and Muon (see Algorithm 2 in the next section) in the stochastic zero-order setting (Sun et al., 2023), as it enables more careful handling of variance $\sigma$ in the gradient estimates (2). Another important difference from prior works is that the result of Lemma 3.4 does not involve a term proportional to $\|\nabla f(x^t)\|_2^2$, which typically appears in analyses where the zero-order estimator (2) is constructed using random uniform or Gaussian vectors $e$ (Cai et al., 2021; Kozak et al., 2021; Gorbunov et al., 2022; Qian & Zhao, 2023). In such cases the deviation $\|m_t - \nabla f(x^t)\|_2$ usually depends on $\|\nabla f(x^t)\|_2$, which substantially complicates proving convergence in terms of $\|\nabla f(x^t)\|$ in the non-convex stochastic zero-order setting. In contrast, Lemma 3.4 shows that for the JAGUAR estimator this deviation is controlled by a noise-dependent constant, which makes the convergence analysis of SignSGD (Algorithm 1) significantly simpler. Table 2 includes a baseline with standard ZO SignSGD based on Gaussian directions, which performs significantly worse than our JAGUAR-based methods, empirically supporting this theoretical distinction. It is worth noting that a similar to Lemma 3.4 result can be obtained when using a full coordinate estimator (Lian et al., 2016). However, this approach requires $\mathcal{O}(d)$ calls to the zero-order oracle per iteration, which can be computationally expensive. In contrast, the JAGUAR method achieves the same result with only $\mathcal{O}(1)$ oracle calls and with the same number of parameters, offering significant improvements in efficiency. This makes our approach particularly attractive for large-scale optimization tasks, where reducing oracle complexity is critical. In Appendix A, we provide an ablation study on $\beta$ and show that the `Jaguar SignSGD` method perform poorly for small $\beta$, while achieving robust high performance around $\beta \approx 0.9$.

With the help of Lemma 3.4, we provide convergence analysis of Algorithm 1.

**Theorem 3.5.** *Consider Assumptions 3.1, 3.2 and 3.3. Then `JAGUAR SignSGD` (Algorithm 1) has the following convergence rate:*

$$\mathbb{E}\left[\left\|\nabla f\left(x^{N(T)}\right)\right\|_1\right] = \mathcal{O}\left[\frac{\delta_0}{\gamma T} + \frac{d\left\|\nabla f(x^0)\right\|_2}{T\sqrt{1-\beta}} + \frac{d^2 L\gamma}{1-\beta} + \sqrt{1-\beta}d\sigma + dL\tau + \frac{d\Delta}{\tau}\right],$$

*where we used a notation $\delta_0 := f(x^0) - f^*$.*

**Corollary 3.6.** *Consider the conditions of Theorem 3.5. In order to achieve the $\varepsilon$-approximate solution (in terms of $\mathbb{E}\left[\left\|\nabla f(x^{N(T)})\right\|_1\right] \leq \varepsilon$), Algorithm 1 needs $T$ iterations (ZO oracle calls), for:*

***Optimal tuning:*** $\gamma = \sqrt{\frac{\delta_0(1-\beta)}{d^2 LT}}$, $\beta = 1 - \min\left\{1; \sqrt{\frac{L\delta_0}{T\sigma^2}}\right\}$, $\tau = (\Delta/L)^{1/2}$ *and* $\varepsilon \geq d\sqrt{\Delta L}$:

$$T = \mathcal{O}\left[\frac{\delta_0 L d^2}{\varepsilon^2} + \frac{\delta_0 L d^2}{\varepsilon^2} \cdot \left(\frac{d\sigma}{\varepsilon}\right)^2\right].$$

**Discussion.** The convergence rate established in Theorem 3.5 is similar to what is known for first-order methods (Bernstein et al., 2018; Jin et al., 2020; Safaryan & Richtárik, 2021; Kornilov et al., 2025), however our bounds include an additional factor of $d$, which is typical for all coordinate-based methods (Nesterov, 2012; Richtárik & Takáč, 2016), not just zero-order ones. This dependence on the dimension arises because coordinate methods process one direction at a time, accumulating complexity proportional to $d$. It is also important to note that without momentum ($\beta = 0$), the

algorithm can only guarantee convergence to a neighbourhood of the optimum of size proportional to $\sigma$, as shown in previous works on zero-order SignSGD (Liu et al., 2019a; Kornilov et al., 2025). Note, that the estimate $T = \mathcal{O}(\varepsilon^{-4})$ in Corollary 3.6 is a lower bound for the non-convex setting and cannot be improved (Arjevani et al., 2023). Let us also point out that we cannot choose an arbitrary $\varepsilon$ in Corollary 3.6, since there exists an irreducible (Dvurechensky et al., 2021; Veprikov et al., 2024) error $\Delta$ in the zero-order oracle (see Assumption 3.3). However, since $\Delta$ is very small, we can still achieve an acceptable accuracy $\varepsilon$. In our analysis, we use the $\ell_1$-norm of the gradient as the convergence criterion, while the standard in non-convex optimization is the $\ell_2$-norm (Euclidean) (Ghadimi & Lan, 2013; 2016). By setting $\varepsilon_{\ell_1} = \sqrt{d} \cdot \varepsilon_{\ell_2}$, we can rescale our result of Corollary 3.6 as

$$T_{\text{Euclidean}} = \mathcal{O}\Big[\frac{\delta_0 L d}{\varepsilon^2} + \frac{\delta_0 L d}{\varepsilon^2} \cdot \Big(\frac{\sqrt{d}\sigma}{\varepsilon}\Big)^2\Big].$$

This substitution allows us to obtain improved results in terms of the dependence on $d$.

### 3.3 ZERO-ORDER MUON WITH JAGUAR GRADIENT APPROXIMATION

In this section, we address the matrix optimization setting, where the optimization variables $X_t$ are elements of the matrix space $\mathbb{R}^{m \times n}$, rather than the standard vector space $\mathbb{R}^d$. Such a formulation allows for a more direct representation of model parameters, helping to better capture their underlying structure (Bernstein & Newhouse, 2024b; Pethick et al., 2025). For the first time in the literature, we introduce a zero-order version of the Muon (Jordan et al., 2024) algorithm (Algorithm 2), broadening the applicability to matrix-structured optimization tasks where only function evaluations are available.

---

**Algorithm 2** Zero-Order Muon with JAGUAR (`JAGUAR Muon`)

1: **Parameters:** stepsize $\gamma$, momentum $\beta$, smoothing parameter $\tau$, number of Newton-Schulz steps ns_steps, number of iterations $T$.
2: **Initialization:** choose $X^0 \in \mathbb{R}^{m \times n}$ and $M^{-1} = \mathbf{0} \in \mathbb{R}^{m \times n}$.
3: **for** $t = 0, 1, 2, \ldots, T$ **do**
4:     Sample $i_t \sim \text{Uniform}(\overline{1,m})$ and $j_t \sim \text{Uniform}(\overline{1,n})$
5:     Set one-hot matrix $E^t$ with 1 in the $(i_t, j_t)$ coordinate
6:     Sample stochastic variable $\xi^t \sim \mathcal{D}$
7:     Set $\widetilde{\nabla}_{i_t j_t} f(X^t, \xi^t) := \frac{\hat{f}(X^t + \tau E^t, \xi^t) - \hat{f}(X^t - \tau E^t, \xi^t)}{2\tau} \in \mathbb{R}$
8:     Set $M^t_{i_t, j_t} = \beta M^{t-1}_{i_t, j_t} + (1 - \beta)\widetilde{\nabla}_{i_t j_t} f(X^t, \xi^t)$ and $M^t_{i \neq i_t, j \neq j_t} = M^{t-1}_{i \neq i_t, j \neq j_t}$
9:     Set $X^{t+1} = X^t - \gamma \cdot \texttt{Newton\_Schulz}(M^t, \text{ns\_steps})$
10: **end for**
11: **Return:** $X^{N(T)}$, where $N(T) \sim \text{Uniform}(\overline{1,T})$.
1: **Subroutine** $\texttt{Newton\_Schulz}(A \in \mathbb{R}^{m \times n}, K = 5)$:
2:     Set $A^0 = A/\|A\|_F$
3:     **for** $k = 0, 1, 2, \ldots, K - 1$ **do**
4:         $A^{k+1} = 3/2 \cdot A^k - 1/2 \cdot A^k (A^k)^T A^k$
5:     **end for**
6:     **Return:** $A^K \approx U_A \cdot V_A^T$.

---

`Newton_Schulz` is an iterative process for matrix orthogonalization (Bernstein & Newhouse, 2024b). Its iteration replaces update matrix with the closest semi-orthogonal matrix to it. This is equivalent to replacing the matrix $A$ by $UV^T$, where $A = U\Sigma V^T$ is its SVD. We choose number of iterations $K = 5$ the same as in (Jordan et al., 2024; Liu et al., 2025). We empirically find this value to be optimal in terms of methods efficiency and its overall performance.

Algorithm 2 is similar to the first-order Muon algorithm (Jordan et al., 2024), the only difference is that we use zero-order gradient approximation JAGUAR (Veprikov et al., 2024) in line 9.

Let us note that when extending to matrix-valued parameters, it is necessary to slightly modify Assumptions 3.1 and 3.2: all occurrences of the $\ell_2$ norm $\| \cdot \|_2$ should be replaced with the Frobenius norm $\| \cdot \|_F$. This modification is justified, as the following property holds for all matrices $A \in \mathbb{R}^{m \times n}$: $\|A\|_F = \|\overline{\text{vec}}(A)\|_2$. We now provide the convergence analysis of `JAGUAR Muon` (Algorithm 2).

**Theorem 3.7.** *Consider Assumptions 3.1, 3.2 (with Frobenius norm) and 3.3. Then* `JAGUAR Muon` *(Algorithm 2) has the following convergence rate:*

$$\mathbb{E}\left[\left\|\nabla f\left(X^{N(T)}\right)\right\|_{\mathcal{S}_1}\right] = \mathcal{O}\left[\frac{\delta_0}{\gamma T} + m^{1/2}n\left(\frac{\|\nabla f(X^0)\|_2}{T\sqrt{1-\beta}} + \frac{mn\gamma}{1-\beta} + \sqrt{1-\beta}\sigma + L\tau + \frac{\Delta}{\tau}\right)\right],$$

*where we used a notation $\delta_0 := f(x^0) - f^*$. We also assume that $n \leq m$.*

**Corollary 3.8.** *Consider the conditions of Theorem 3.7. In order to achieve the $\varepsilon$-approximate solution (in terms of $\mathbb{E}[\|\nabla f(X^{N(T)})\|_{\mathcal{S}_1}] \leq \varepsilon$), Algorithm 2 needs $T$ iterations (ZO calls), for:*

**Optimal tuning:** $\gamma = \sqrt{\frac{\delta_0(1-\beta)}{m^{3/2}n^2LT}}, \beta = 1 - \min\left\{1; \sqrt{\frac{L\delta_0}{T\sigma^2}}\right\}, \tau = (\Delta/L)^{1/2}, \varepsilon \geq m^{1/2}n\sqrt{\Delta L}$:

$$T = \mathcal{O}\left[\frac{\delta_0 Lm^{3/2}n^2}{\varepsilon^2} + \frac{\delta_0 Lm^{3/2}n^2}{\varepsilon^2} \cdot \left(\frac{m^{3/2}n^2\sigma}{\varepsilon}\right)^2\right].$$

**Discussion.** The convergence rate established in Theorem 3.7 is consistent with the first-order case (Li & Hong, 2025; Kovalev, 2025). However, there remain zero-order terms depending on $\tau$ and $\Delta$, as for Algorithm 1 (see Theorem 3.5 and Discussion part after it). From a proof perspective, Theorems 3.5 and 3.7 are very similar, since the orthogonalization operation (`Newton_Schulz`) in Algorithm 2 can be interpreted as taking the sign of the gradient matrix eigenvalues. Accordingly, both the form and the convergence rate criterion are analogous (the $\ell_1$ norm for Algorithm 1 and the $\mathcal{S}_1$ norm for Algorithm 2). Nevertheless, the convergence rates of the two algorithms differ slightly. We examine the two boundary cases in the following remark.

*Remark* 3.9. For optimal tuning from Corollary 3.8 we can specify the number of iterations of Algorithm 2 to achieve the $\varepsilon$-approximate solution in terms of the total number of parameters $d = m \cdot n$ in the two boundary cases:

- If $n \ll m \approx d$: $\quad T_{n\ll m\approx d} = \mathcal{O}\left[\frac{\delta_0 Ld^{3/2}}{\varepsilon^2} + \frac{\delta_0 Ld^{3/2}}{\varepsilon^2} \cdot \left(\frac{d^{3/2}\sigma}{\varepsilon}\right)^2\right].$

- If $n \approx m \approx \sqrt{d}$: $\quad T_{n\approx m\approx\sqrt{d}} = \mathcal{O}\left[\frac{\delta_0 Ld^{7/4}}{\varepsilon^2} + \frac{\delta_0 Ld^{7/4}}{\varepsilon^2} \cdot \left(\frac{d^{7/4}\sigma}{\varepsilon}\right)^2\right].$

Accordingly, comparing these convergence rates with that obtained in Corollary 3.8, we observe an improvement by factors of $d^{1/2}$ and $d^{1/4}$, respectively.

## 4    EXPERIMENTS

In this section, we empirically evaluate our proposed ZO optimization methods for fine-tuning large language models, focusing on both accuracy and memory efficiency. Building on the framework of (Zhang et al., 2024b), we extend the evaluation to include `JAGUAR SignSGD` (Algorithm 1) and `JAGUAR Muon` (Algorithm 2), aiming to achieve competitive downstream accuracy with baseline-level memory usage. We also introduce `ZO-Muon` (Algorithm 3, Appendix B), a zero-order adaptation of Muon (Jordan et al., 2024) based on Gaussian gradient estimation (2).

Table 2: Test accuracy on SST2 for OPT-1.3B and RoBERTa-Large with FT and LoRA. Best performance among ZO methods is in **bold**.

| Method | OPT-1.3B | | RoBERTa-Large | |
|---|---|---|---|---|
| | FT | LoRA | FT | LoRA |
| FO-SGD | 91.1 | 93.6 | 91.4 | 91.2 |
| FO-Muon | 87.3 | - | 88.4 | - |
| Forward-Grad | 90.3 | 90.3 | 90.1 | 89.7 |
| ZO-SGD | 90.8 | 90.1 | 89.4 | 90.8 |
| Acc-ZOM | 85.2 | 91.3 | 89.6 | 90.9 |
| ZO-SGD-Cons | 88.3 | 90.5 | 89.6 | 91.6 |
| ZO-SignSGD | 87.2 | 91.5 | 52.5 | 90.2 |
| ZO-AdaMM | 84.4 | 92.3 | 89.8 | 89.5 |
| LeZO | 85.1 | 92.3 | 90.4 | 91.8 |
| `JAGUAR SignSGD` | **94.0 ± 0.1** | 92.5 ± 0.5 | **92.2 ± 0.2** | **92.2 ± 0.4** |
| `JAGUAR Muon` | 86.0 ± 0.1 | **94.0 ± 0.1** | 85.0 ± 0.1 | **92.2 ± 0.2** |
| `ZO-Muon` | 86.5 ± 0.1 | 93.5 ± 0.1 | 72.0 ± 0.1 | 86.0 ± 0.2 |

### 4.1    EXPERIMENTAL SETUP

**Fine-Tuning Task and Schemes.** Fine-tuning LLMs is a pivotal process in adapting pre-trained models to downstream tasks, enabling high performance with limited task-specific data. To explore

the efficacy of our ZO methods, we follow the recent ZO fine-tuning benchmark of (Zhang et al., 2024b). Concretely, we consider the SST2 dataset (Socher et al., 2013), the WinoGrande (Sakaguchi et al., 2021) and COPA (Roemmele et al., 2011) datasets, a widely used benchmarks for LLM fine-tuning (Zhang et al., 2024b; Chen et al., 2024; Malladi et al., 2023). To further assess generalization, we include the HumanEval code-generation benchmark (Chen, 2021), which is substantially more challenging: it requires program synthesis and functional correctness, and is widely regarded as a demanding test of LLMs' reasoning and generation capabilities.

We consider two fine-tuning schemes:

- **Full Fine-Tuning (FT):** Updates all parameters of the pre-trained model.

- **Low-Rank Adaptation (LoRA) (Hu et al., 2021):** Introduces a small set of trainable parameters while keeping the original model parameters frozen, which reduces memory compared to full FT but still requires full backpropagation and thus significantly higher memory than ZO methods (see updated memory comparison in Table 4)

**Models.** We conduct experiments using four prominent LLMs: OPT-1.3B (Zhang et al., 2022), a 1.3 billion parameter model from the OPT family; RoBERTa-Large (Liu et al., 2019b), a 355 million parameter model known for its robust performance in natural language processing tasks. For more challenging benchmarks we utilize Llama 2 (Touvron et al., 2023), OPT-13B (Zhang et al., 2022) and Gemma3-7B (Kamath et al., 2025), state-of-the-art open-source models widely used for research and applications, designed for strong generative performance and reasoning-heavy benchmarks. These models represent a range of sizes and architectures, allowing us to assess the scalability and generality of our methods.

Table 3: Test accuracy on COPA and WinoGrande for OPT-13B and Llama2-7B with LoRA. Best performance among ZO methods is in **bold**.

| Method | OPT-13B | | LLaMA2-7B | |
|---|---|---|---|---|
| | COPA | WinoGrande | COPA | WinoGrande |
| FO-SGD | 88 | 66.9 | 85 | 66.9 |
| Forward-Grad | 89 | 62.9 | 82 | 64.3 |
| ZO-SGD | 87 | 62.6 | 86 | 64.3 |
| ZO-SGD-Cons | 88 | 63.3 | 85 | 64.6 |
| JAGUAR SignSGD | **89 ± 0.3** | **63.7 ± 0.1** | **88 ± 0.2** | **64.9 ± 0.1** |
| JAGUAR Muon | 87 ± 0.2 | 62.3 ± 0.2 | **88 ± 0.1** | 62.8 ± 0.2 |
| ZO-Muon | 87 ± 0.2 | 61.9 ± 0.3 | 85 ± 0.2 | 61.6 ± 0.2 |

Table 4: GPU allocated memory (GB) for OPT-13B and LLaMA2-7B on WinoGrande and COPA with LoRA

| Method | OPT-13B | | LLaMA2-7B | |
|---|---|---|---|---|
| | COPA | WinoGrande | COPA | WinoGrande |
| FO-SGD | 96.247 | 97.355 | 48.572 | 49.114 |
| ZO-SGD | 24.710 | 26.407 | 13.219 | 14.670 |
| ZO-Adam | 38.612 | 39.872 | 27.971 | 29.440 |
| JAGUAR SignSGD | 24.712 | 26.408 | 13.219 | 14.672 |
| JAGUAR Muon | 25.880 | 27.440 | 16.032 | 17.992 |
| ZO-Muon | 25.740 | 27.416 | 15.021 | 16.992 |

**Methods.** We evaluate the following ZO optimization methods proposed in this work:

- **JAGUAR SignSGD:** Combines the JAGUAR gradient approximation with SignSGD and momentum for efficient updates (Algorithm 1).

- **JAGUAR Muon:** Integrates JAGUAR with the Muon optimizer, incorporating momentum and orthogonalization (Algorithm 2).

- **ZO-Muon:** A novel ZO adaptation of the Muon optimizer, leveraging matrix-based optimization principles (Algorithm 3 in Appendix B).

**Comparison procedure.** For comparison, we include baseline methods from (Zhang et al., 2024b), including ZO-SGD (Ghadimi & Lan, 2013), Acc-ZOM (Huang et al., 2022), ZO-SGD-Cons (Kim et al., 2025), ZO-SignSGD (Liu et al., 2019a), ZO-AdaMM (Chen et al., 2019), Forward-Grad (Baydin et al., 2022), and FO-SGD (Amari, 1993), with results reported in the benchmark. We also include LeZO (Wang et al., 2024), which uses a layer-wise selection similar to JAGUAR SignSGD. Experiments for our methods follow the setup of (Zhang et al., 2024b).

## 4.2 RESULTS

**OPT-1.3B and RoBERTa-Large models.** Table 2 reports SST2 test accuracy for OPT-1.3B and RoBERTa-Large under different fine-tuning schemes. Our methods generally outperform baseline ZO approaches. In particular, Algorithms 1 and 2, which employ the JAGUAR gradient approximation, surpass methods relying on random vector sampling ((2)) or FO-style momentum. However, ZO-Muon and JAGUAR Muon exhibit weaker FT performance, likely due to non-matrix parameters in full FT.

**OPT-13B and Llama2-7B models.** We additionally conduct experiments with large-size models: OPT-13B (Zhang et al., 2022) and Llama2-7B (Touvron et al., 2023) on WinoGrande (Sakaguchi et al., 2021) and COPA (Roemmele et al., 2011) tasks. We use cosine scheduler for Llama2-7B and polynomial decay for OPT-13B. We repeat the evaluation results from (Zhang et al., 2024b) as baselines in Table 3. However, the cited work does not report memory efficiency, a key metric in parameter-efficient fine-tuning. We excluded ZO-AdaMM due to its excessive memory use.

**Gemma-7B and HumanEval.** We further evaluate our methods on the Gemma-7B model fine-tuned on the HumanEval benchmark, which focuses on code generation and functional correctness. Across this setting, `JAGUAR SignSGD` and `JAGUAR Muon` achieve competitive or superior pass@1 (measured as the percentage of successfully passed unit-tests run with generated code on the test set) performance compared to ZO-SGD and other baselines, confirming that the proposed momentum-based ZO methods transfer effectively to challenging generative and reasoning-heavy tasks.

Table 5: Pass@1, maximum memory consumption, and wallclock time on HumanEval for Gemma-7B in full FT scheme. Best performance among ZO methods is in **bold**.

| Method | Pass@1 | Memory (GB) | Seconds per step |
|---|---|---|---|
| Baseline (no FT) | 0.51 | - | - |
| FO-SGD | 0.86 | 108.46 | 4.09 |
| ZO-SGD | 0.61 | **73.03** | 3.61 |
| ZO-SignSGD | 0.64 | **73.03** | 3.60 |
| `JAGUAR SignSGD` | 0.67 | 75.37 | **3.27** |
| `JAGUAR Muon` | **0.74** | 75.39 | 3.69 |
| `ZO-Muon` | 0.63 | 73.22 | 3.94 |

**Discussion.** Tables 2 and 3 show that `JAGUAR SignSGD` and `JAGUAR Muon` outperform baselines, confirming their effectiveness and robustness. Our methods remain scalable and practical, particularly in memory-constrained, high-capacity settings.

The results from Table 5 highlight a clear accuracy–efficiency trade-off between first-order and zero-order optimizing algorithms, and show that the proposed JAGUAR variants substantially narrow the gap to FO under tight resource budgets. As expected, FO-SGD attains the highest Pass@1 but at a steep memory cost and slower step time. In contrast, standard ZO baselines reduce peak memory by roughly 30–35 GB and modestly improve step time, but with lower Pass@1. The JAGUAR methods improve on these ZO baselines: `JAGUAR Muon` achieves the best performance (0.74) among ZO methods, cutting the FO quality gap by more than half while keeping memory consumption sufficiently low. Notably, `ZO-Muon` variant underperforms `JAGUAR Muon`, suggesting that JAGUAR's design choices drive the gains.

**Memory Efficiency.** Table 4 compares GPU allocated memory for Llama2-7B and OPT-13B highlighting the efficiency of our methods. Results of this experiment demonstrate that our approaches effectively balance accuracy gains with memory efficiency. Note that memory consumption from Table 4 and Table 5 differs significantly even for FO optimization algorithms despite comparable datasets and models sizes. The effect of growing memory consumption is caused by the types of the tasks: generation benchmark requires generation of many more tokens than classification or single-word answers. What is more, code-generation specifically requires storing canonical solution and unit tests in order to evaluate Pass@k performance.

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

# A  ABLATION STUDIES

## A.1  ABLATIONS ON MOMENTUM AND SMOOTHING PARAMETER

We present an ablation study on the effect of the $\beta$ parameter from Algorithm 1 on learning efficiency. Figure 1 reports the accuracy of the `JAGUAR SignSGD` method on the SST-2 dataset with the RoBERTa-large model across different values of $\beta$. The results indicate that the choice of $\beta$ has a significant impact on model performance. Specifically, small values of $\beta$ lead to substantially lower accuracy, suggesting that insufficient momentum or smoothing in the update steps can hinder effective learning. As $\beta$ increases, the method benefits from more stable gradient aggregation, resulting in improved convergence behavior. Notably, around $\beta \approx 0.9$, the method achieves robust and consistently high performance, indicating that this range provides an optimal balance between responsiveness to new gradient information and stability in updates. This highlights the importance of tuning $\beta$ carefully to maximize the learning efficiency and predictive performance of `JAGUAR SignSGD`.

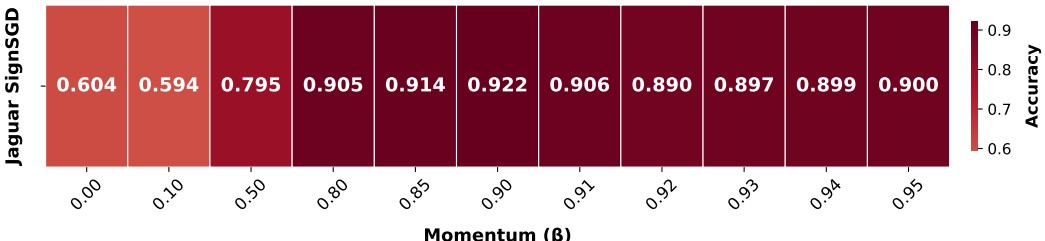

Figure 1: Test accuracy of `JAGUAR SignSGD` on SST-2 RoBERTa-large with LoRA for different values of $\beta$.

We additionally present an ablation study on smoothing parameter $\tau$ for the `JAGUAR SignSGD` method from Algorithm 1 in the same setup discussed above. We show that there is no strict dependence on the $\tau$ value. Figure 2 represents that it's important to have $\tau \leq 1$, setting it around $10^{-4}$. The method remains robust for $\tau$ in the range from $10^{-4}$ to $5 \times 10^{-3}$, with consistently strong performance. Beyond this range, however, accuracy begins to degrade, indicating that larger smoothing values introduce instability and lead to noticeably worse results.

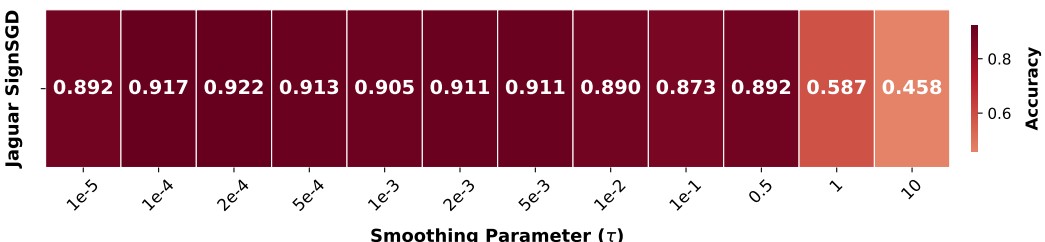

Figure 2: Test accuracy of `JAGUAR SignSGD` on SST-2 RoBERTa-large with LoRA for different values of $\tau$.

## A.2  ABLATION ON NUMBER OF PERTURBED LAYERS

We further investigate the influence of the number of model layers perturbed during each update step on the performance of `JAGUAR Muon` in the RoBERTa-large LoRA setting for the SST-2 dataset. This value controls the sparsity of the perturbation mask and therefore determines the extent to which the algorithm explores the parameter space during zero-order gradient estimation.

We perturb entire layers rather than individual scalar coordinates because the optimization loop naturally operates at the layer level. As a result, perturbing one layer has essentially the same time and memory cost as perturbing a single coordinate, while aligning with how parameters are accessed and updated in practice.

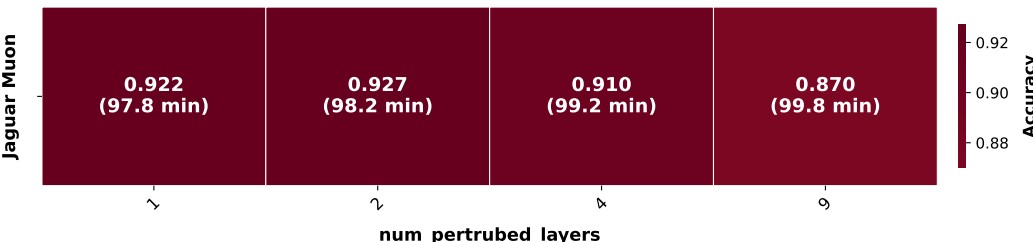

Figure 3: Test accuracy and wall clock time of `JAGUAR Muon` on SST-2 for RoBERTa-large with LoRA for different values of `num_pertrubed_layers`.

The results in Figure 3 show a non-monotonic dependence of downstream accuracy on the number of perturbed layers. This makes sense if we recall that a zero-order update estimates the gradient only along one sampled direction. Changing `num_perturbed_layers` does not change this direction, but only how many coordinates in it are non-zero, therefore increasing the number of perturbed layers makes this direction less sparse and can weaken the useful signal. In our experiments, perturbing one or two layers gives the best performance, while perturbing four or nine layers leads to a stable drop. Also its worth noticing, that increasing `num_perturbed_layers` naturally incurs a higher computational cost per step, our measurements show that the resulting increase in wall-clock time is negligible.

## B    CLASSICAL ZO MUON

Using gradient estimate in the form (2), we adapt the Muon algorithm (Jordan et al., 2024) into zero-order form:

---

**Algorithm 3** Zero-Order Muon (`ZO-Muon`)

---

1: **Parameters:** stepsize (learning rate) $\gamma$, gradient approximation parameter $\tau$, number of iterations $T$.
2: **Initialization:** choose $X^0 \in \mathbb{R}^{m \times n}$
3: **for** $t = 0, 1, 2, \ldots, T$ **do**
4:      Sample $E^t \in \mathbb{R}^{m \times n}$ from $\mathcal{N}(0, 1)$
5:      Compute $G^t = \frac{\hat{f}(X^t + \tau E^t) - \hat{f}(X^t - \tau E^t)}{2\tau} E^t$
6:      Set $X^{t+1} = X^t - \gamma \cdot \texttt{Newton\_Schulz}(G^t)$
7: **end for**
8: **Return:** $X^T$

---

## C  ADDITIONAL EXPERIMENTS AND FINE-TUNING SETUP

### C.1  CONVERGENCE WITH RESPECT TO WALL-CLOCK TIME

In this section, we report additional convergence results that evaluate the practical efficiency of the considered zero-order methods in terms of wall-clock time. Figure 4 shows representative training curves where the horizontal axis corresponds to wall-clock time. We plot accuracy on test dataset for all ZO baselines, FO-SGD and the proposed JAGUAR-based methods (Algorithms 1 and 2) under the same experimental setup as in the main text (e.g., Table 2). It shows that ZO methods require less wall-clock time to reach competitive accuracy, demonstrating that they offer a favorable speed–efficiency trade-off and can outperform FO methods in practical convergence time despite relying solely on function evaluations. Notably, `JAGUAR SignSGD` (Algorithm 1) converges faster than standard `ZO-SignSGD`, indicating that when perturbing only one layer direction, the JAGUAR update can reduce computational time while simultaneously achieving better accuracy. This results can be generalized for all classification tasks (Zhang et al., 2024b) used in this paper. For generation task we observe the same trend (see accuracy-vs-time analysis analysis in Table 5).

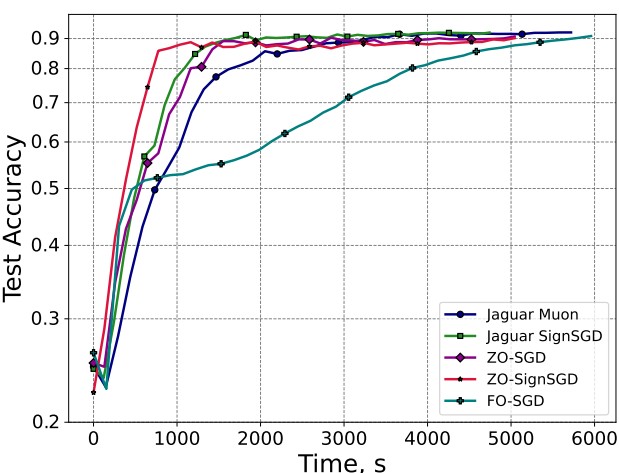

Figure 4: Accuracy-vs-time analysis for all ZO baselines, FO-SGD and the proposed methods on SST-2 for RoBERTa-large with LoRA.

### C.2  EVALUATION PROCEDURE

**Schedulers.** We conducted experiments with different scheduling types. Therefore, results for `Jaguar SignSGD` (Algorithm 1), `Jaguar Muon` (Algorithm 2), and `ZO-Muon` (Algorithm 3) from Tables 2, 3, and 5 are obtained using polynomial or cosine scheduling technique.

**Hyperparameter Tuning.** To ensure optimal performance, we conducted a grid search over key hyperparameters for each method:

- Momentum parameter: $\beta \in \{10^{-3}, 10^{-2}, 10^{-1}, 8 \cdot 10^{-1}\}$,
- Learning rate: $\gamma \in [10^{-6}, 10^{-1}]$,
- Smoothing parameter: $\tau \in \{10^{-1}, 10^{-2}, 10^{-3}, 10^{-4}\}$.

Additional fixed parameters include an epsilon of $10^{-3}$ for numerical stability. The best-performing hyperparameters for each algorithm are detailed on our github `https://anonymous.4open.science/r/zo_jaguar`.

**Evaluation Metrics.** We assess performance using:

- **Test Accuracy:** Measured as the percentage of correct predictions on the test set, reflecting model effectiveness.

- **Pass@k:** Measured as the percentage of successfully passed unit-tests run with generated code on the test set. For each unit-test there are $k$ generated sequences, if any of these passes the test, then this example is considered positive.

- **GPU allocated memory:** Quantified in gigabytes (GB) during training, indicating memory efficiency.

**Implementation Details.** Experiments were conducted with three independent runs per configuration, each with a randomly selected seed fixed at the start to ensure reproducibility. We report the mean and standard deviation of test accuracy in Tables 2 and 3. Following (Malladi et al., 2023), we employed half-precision (F16) training for ZO methods and mixed-precision (FP16) training for FO methods to optimize memory usage. We use LoRA (Hu et al., 2021) fine-tuning strategy with $r = 16$. Training was performed on a single NVIDIA A100 GPU and a single NVIDIA H100 GPU, with memory profiling conducted using standard PyTorch utilities.

Due to HumanEval benchmark does have pre-defined train-test split, we randomly assigned 50 examples to the test set and the rest 114 examples to train sets. Due to the small size of the dataset fine-tuning scheme was held with a low number of epochs to prevent overfitting.

## C.3 EXPERIMENTAL METHODOLOGY

Our experimental procedure was designed to rigorously evaluate the proposed methods under controlled conditions. We consider diffirent datasets (SST2, COPA, WinoGrande, HumanEval), models (OPT-1.3B, RoBERTa-Large, Llama2 7B, OPT-13B, Gemma3-7B), fine-tuning schemes (FT, LoRA), and ZO and FO optimization methods (see Tables 2 and 3). We executed the following steps:

1. **Initialization:** Loaded the pre-trained model and initialized trainable parameters (all for FT, LoRA-specific for LoRA).

2. **Hyperparameter Selection:** Performed a preliminary parameter search to identify the best hyperparameters per method, iterating over the specified ranges and selecting based on validation accuracy.

3. **Evaluation:** Computed test accuracy on the dataset test set after each run, averaging results across three runs with different seeds.

4. **Memory Profiling:** Recorded GPU allocated memory during training, ensuring consistency by maintaining identical hardware settings.

This methodology ensures a fair comparison across methods, capturing both performance and resource utilization comprehensively.

## D    PROOFS FOR ZO MOMENTUM SIGNSGD WITH JAGUAR (ALGORITHM 1)

### D.1    PROOF OF LEMMA 3.4

*Proof.* We start with applying one step recursion to the momentum form the Algorithm 1:

$$\mathbb{E}\left[\left\|m^t - \nabla f(x^t)\right\|_2^2\right] = \mathbb{E}\Big[\Big\|m^{t-1} - (1-\beta)\left\langle m^{t-1}, e^t\right\rangle e^t$$

$$+ (1-\beta)\widetilde{\nabla}_{i_t} f(x^t, \xi^t) - \nabla f(x^t)\Big\|_2^2\Big]$$

$$= \mathbb{E}\Big[\Big\|\underbrace{\left\{I - (1-\beta)e^t(e^t)^T\right\}\left\{m^{t-1} - \nabla f(x^{t-1})\right\}}_{=:a^t}$$

$$+ (1-\beta)e^t(e^t)^T\underbrace{\left\{\widetilde{\nabla} f(x^t, \xi^t) - \nabla f(x^t)\right\}}_{=:b^t} \qquad (3)$$

$$- \underbrace{\left\{I - (1-\beta)e^t(e^t)^T\right\}\left\{\nabla f(x^t) - \nabla f(x^{t-1})\right\}}_{=:c^t}\Big\|_2^2\Big],$$

where we used a notation $\widetilde{\nabla} f(x, \xi) := \sum_{i=1}^d \frac{\hat{f}(x+\tau e^i, \xi) - \hat{f}(x-\tau e^i, \xi)}{2\tau} e^i$, and $e^i$ is the one-hot vector with 1 in the $i$-th coordinate. In equation (3) we also used the classical notation of the identity matrix $I \in \mathbb{R}^{d \times d}$.

Now using axillary notations $a^t, b^t, c^t$ from equation (3), we divide it into six parts:

$$\mathbb{E}\left[\left\|a^{t+1}\right\|_2^2\right] = \underbrace{\mathbb{E}\left[\left\|\left\{I - (1-\beta)e^t(e^t)^T\right\}a^t\right\|_2^2\right]}_{①}$$

$$+ \underbrace{\mathbb{E}\left[\left\|(1-\beta)e^t(e^t)^T b^t\right\|_2^2\right]}_{②}$$

$$+ \underbrace{\mathbb{E}\left[\left\|\left\{I - (1-\beta)e^t(e^t)^T\right\}c^t\right\|_2^2\right]}_{③} \qquad (4)$$

$$+ \underbrace{\mathbb{E}\left[2\left\langle\left\{I - (1-\beta)e^t(e^t)^T\right\}a^t, (1-\beta)e^t(e^t)^T b^t\right\rangle\right]}_{④}$$

$$- \underbrace{\mathbb{E}\left[2\left\langle\left\{I - (1-\beta)e^t(e^t)^T\right\}a^t, \left\{I - (1-\beta)e^t(e^t)^T\right\}c^t\right\rangle\right]}_{⑤}$$

$$- \underbrace{\mathbb{E}\left[2\left\langle(1-\beta)e^t(e^t)^T b^t, \left\{I - (1-\beta)e^t(e^t)^T\right\}c^t\right\rangle\right]}_{⑥}.$$

Consider ①. Since $i_t$ from Algorithm 1 is generated independent and uniform and $\{m^{s-1}, x^s\}_{s=0}^t$ do not depend on $i_t$, we can apply tower property:

$$① = \mathbb{E}\left[\left\|\left\{I - (1-\beta)e^t(e^t)^T\right\}a^t\right\|_2^2\right]$$

$$= \mathbb{E}\left[(a^t)^T\left\{I - (1-\beta)e^t(e^t)^T\right\}^T\left\{I - (1-\beta)e^t(e^t)^T\right\}a^t\right]$$

$$= \mathbb{E}\left[(a^t)^T\left\{I - (1-\beta)(2-(1-\beta))e^t(e^t)^T\right\}a^t\right]$$

$$= \mathbb{E}\left[(a^t)^T \cdot \mathbb{E}_{i_t \sim \text{Uniform}(\overline{1,d})}\left[I - (1-\beta^2)e^t(e^t)^T\right] \cdot a^t\right]$$

$$= \mathbb{E}\left[(a^t)^T \cdot \left(1 - \frac{1-\beta^2}{d}\right)I \cdot a^t\right] = \left(1 - \frac{1-\beta^2}{d}\right)\mathbb{E}\left[\left\|a^t\right\|_2^2\right]. \qquad (5)$$

Here we used the fact that $\left(e^t(e^t)^T\right)^T e^t(e^t)^T = e^t(e^t)^T$ and $\mathbb{E}_{i_t \sim \text{Uniform}(\overline{1,d})}\left[e^t(e^t)^T\right] = \frac{1}{d}I$.

Similarly to equation (5), we can estimate ② and ③:

$$② = \mathbb{E}\left[\left\|(1-\beta)e^t(e^t)^T b^t\right\|_2^2\right] = \frac{(1-\beta)^2}{d}\mathbb{E}\left[\left\|b^t\right\|^2\right],$$

$$③ = \mathbb{E}\left[\left\|\left\{I-(1-\beta)e^t(e^t)^T\right\}c^t\right\|_2^2\right] = \left(1-\frac{1-\beta^2}{d}\right)\mathbb{E}\left[\left\|c^t\right\|^2\right].$$

Since $b^t = \widetilde{\nabla}f(x^t, \xi^t) - \nabla f(x^t)$, we can use Lemma 4 from (Veprikov et al., 2024) with $\sigma_f = 0, \sigma_\nabla = \sigma$ and obtain the result of the form:

$$② \leq \frac{(1-\beta)^2}{d}\cdot\left(dL^2\tau^2 + 2d\sigma^2 + \frac{2d\Delta^2}{\tau^2}\right), \tag{6}$$

where $L, \sigma$ and $\Delta$ come from Assumptions 3.1, 3.2 and 3.3.

Since $c^t = \nabla f(x^t) - \nabla f(x^{t-1})$, we can use Assumption 3.1 and obtain:

$$③ \leq \left(1-\frac{1-\beta^2}{d}\right)L^2\left\|x^t - x^{t-1}\right\|_2^2 = \left(1-\frac{1-\beta^2}{d}\right)L^2\left\|\mathrm{sign}(m^t)\right\|_2^2$$

$$= \left(1-\frac{1-\beta^2}{d}\right)dL^2\gamma^2 \leq dL^2\gamma^2. \tag{7}$$

Consider ④. Let us move all matrixes to the left side of the dot product:

$$④ = \mathbb{E}\left[2\left\langle(1-\beta)\left\{I-(1-\beta)e^t(e^t)^T\right\}e^t(e^t)^T\cdot a^t, b^t\right\rangle\right]$$

$$= \mathbb{E}\left[2\left\langle(1-\beta)\beta e^t(e^t)^T\cdot a^t, b^t\right\rangle\right].$$

Now we use tower property for $i_t$ as we did for ①, ②, ③ and use the definitions of $a^t$ and $b^t$:

$$④ = \frac{(1-\beta)\beta}{d}\cdot\mathbb{E}\left[2\left\langle a^t, b^t\right\rangle\right]$$

$$= \frac{(1-\beta)\beta}{d}\cdot\mathbb{E}\left[2\left\langle m^{t-1} - \nabla f(x^{t-1}), \widetilde{\nabla}f(x^t,\xi^t) - \nabla f(x^t)\right\rangle\right].$$

We now again use tower property, but with stochastic variable $\xi^t$. Since $\{m^{s-1}, x^s\}_{s=0}^t$ do not depend on $\xi^t$, we can obtain that:

$$④ = \frac{(1-\beta)\beta}{d}\cdot\mathbb{E}\left[2\left\langle m^{t-1} - \nabla f(x^{t-1}), \mathbb{E}_{\xi^t}\left[\widetilde{\nabla}f(x^t,\xi^t)\right] - \nabla f(x^t)\right\rangle\right]$$

$$\leq \frac{(1-\beta)\beta}{2d}\cdot\mathbb{E}\left[\left\|m^{t-1} - \nabla f(x^{t-1})\right\|_2^2\right] \tag{8}$$

$$+ \frac{2(1-\beta)\beta}{d}\cdot\mathbb{E}\left[\left\|\mathbb{E}_{\xi^t}\left[\widetilde{\nabla}f(x^t,\xi^t)\right] - \nabla f(x^t)\right\|_2^2\right].$$

In (8) we use Fenchel-Young inequality. For estimating $\|\mathbb{E}_{\xi^t}[\widetilde{\nabla}f(x^t,\xi^t)] - \nabla f(x^t)\|_2^2$ we again can use Lemma 4 from (Veprikov et al., 2024) but now with $\sigma_\nabla = \sigma_f = 0$ since we have no randomness in $\mathbb{E}_{\xi^t}\left[\widetilde{\nabla}f(x^t,\xi^t)\right]$. Therefore ④ is bounded as:

$$④ \leq \frac{(1-\beta)\beta}{2d}\cdot\mathbb{E}\left[\left\|a^t\right\|_2^2\right] + \frac{2(1-\beta)\beta}{d}\cdot\left(dL^2\tau^2 + \frac{2d\Delta^2}{\tau^2}\right). \tag{9}$$

Consider ⑤. Similar to ④ we can obtain:

$$⑤ = \mathbb{E}\left[2\left\langle\left\{I-(1-\beta)e^t(e^t)^T\right\}a^t, \left\{I-(1-\beta)e^t(e^t)^T\right\}c^t\right\rangle\right]$$

$$= \mathbb{E}\left[2\left\langle\left\{I-(1-\beta^2)e^t(e^t)^T\right\}a^t, c^t\right\rangle\right]$$

$$= \left(1-\frac{1-\beta^2}{d}\right)\cdot\mathbb{E}\left[2\left\langle a^t, c^t\right\rangle\right]$$

$$\leq \left(1-\frac{1-\beta^2}{d}\right)\cdot\frac{1-\beta}{2d}\cdot\mathbb{E}\left[\left\|a^t\right\|_2^2\right] + \left(1-\frac{1-\beta^2}{d}\right)\cdot\frac{2d}{1-\beta}\cdot\mathbb{E}\left[\left\|c^t\right\|_2^2\right]$$

$$\leq \frac{1-\beta}{2d} \cdot \mathbb{E}\left[\left\|a^t\right\|_2^2\right] + \frac{2d}{1-\beta} \cdot dL^2\gamma^2. \tag{10}$$

Finally, we estimate ⑥ in the same way:

$$
\begin{aligned}
⑥ &= \mathbb{E}\left[2\left\langle (1-\beta)e^t(e^t)^T b^t, \left\{I - (1-\beta)e^t(e^t)^T\right\}c^t\right\rangle\right] \\
&= \mathbb{E}\left[2\left\langle (1-\beta)\beta e^t(e^t)^T b^t, c^t\right\rangle\right] \\
&= \frac{(1-\beta)\beta}{d} \cdot \mathbb{E}\left[2\left\langle b^t, c^t\right\rangle\right] \\
&\leq \frac{(1-\beta)\beta}{d} \cdot \mathbb{E}\left[\left\|\mathbb{E}_{\xi^t}\left[\widetilde{\nabla}f(x^t, \xi^t)\right] - \nabla f(x^t)\right\|_2^2\right] + \frac{(1-\beta)\beta}{d} \cdot \mathbb{E}\left[\left\|c^t\right\|_2^2\right] \\
&\leq \frac{(1-\beta)\beta}{d} \cdot \left(dL^2\tau^2 + \frac{2d\Delta^2}{\tau^2}\right) + \frac{(1-\beta)\beta}{d} \cdot dL^2\gamma^2. \tag{11}
\end{aligned}
$$

We made it! Now let us combine equations (5), (6), (7), (9), (10) and (11) to bound $\mathbb{E}[\|a^{t+1}\|_2^2]$ from equation (4):

$$
\begin{aligned}
\mathbb{E}\left[\left\|a^{t+1}\right\|_2^2\right] &\leq \left(1 - \frac{1-\beta}{d}\left[\underbrace{1+\beta}_{(5)} - \underbrace{\frac{\beta}{2}}_{(9)} - \underbrace{\frac{1}{2}}_{(10)}\right]\right) \cdot \mathbb{E}\left[\left\|a^t\right\|_2^2\right] \\
&\quad + \frac{1-\beta}{d}\left(\underbrace{1-\beta}_{(6)} + \underbrace{2\beta}_{(9)} + \underbrace{\beta}_{(11)}\right)\cdot\left(dL^2\tau^2 + \frac{2d\Delta^2}{\tau^2}\right) + \underbrace{\frac{(1-\beta)^2}{d}}_{(6)}\cdot 2d\sigma^2 \\
&\quad + \left(\underbrace{1}_{(7)} + \underbrace{\frac{2d}{1-\beta}}_{(10)} + \underbrace{\frac{(1-\beta)\beta}{d}}_{(11)}\right)\cdot dL^2\gamma^2 \\
&\leq \left(1 - \frac{1-\beta^2}{2d}\right)\cdot\mathbb{E}\left[\left\|a^t\right\|_2^2\right] \\
&\quad + 3\frac{1-\beta}{d}\cdot\left(dL^2\tau^2 + \frac{2d\Delta^2}{\tau^2}\right) + 2\frac{(1-\beta)^2}{d}\cdot d\sigma^2 + \frac{4d}{1-\beta}\cdot dL^2\gamma^2.
\end{aligned}
$$

By unrolling the recursion in the last inequality we obtain:

$$
\begin{aligned}
\mathbb{E}\left[\left\|m^t - \nabla f(x^t)\right\|_2^2\right] &\leq 8\frac{d^2}{(1-\beta)(1-\beta^2)}\cdot dL^2\gamma^2 + 4\frac{(1-\beta)^2}{1-\beta^2}\cdot d\sigma^2 \\
&\quad + 6\frac{1-\beta}{1-\beta^2}\cdot\left(dL^2\tau^2 + \frac{2d\Delta^2}{\tau^2}\right) + \left(1 - \frac{1-\beta^2}{2d}\right)^t\left\|\nabla f(x^0)\right\|_2^2 \\
&= \mathcal{O}\left[\frac{d^3}{(1-\beta)^2}L^2\gamma^2 + (1-\beta)d\sigma^2 + L^2\tau^2 + \frac{d\Delta^2}{\tau^2}\right. \\
&\quad \left. + \left(1 - \frac{1-\beta^2}{2d}\right)^t\left\|\nabla f(x^0)\right\|_2^2\right].
\end{aligned}
$$

This finishes the proof. $\qquad\square$

### D.2 Proof of Theorem 3.5

*Proof.* We start from using Lemma 1 from (Sun et al., 2023). For the points $x^t$, generated by Algorithm 1 it holds that:

$$f(x^{t+1}) - f(x^t) \leq -\gamma\left\|\nabla f(x^t)\right\|_1 + 2\sqrt{d}\gamma\left\|m^t - \nabla f(x^t)\right\|_2 + \frac{dL\gamma^2}{2}. \tag{12}$$

Now we take mathematical expectation of the both sides of the inequality (12) and use the results from Lemma 3.4. Specifically, from Lemma 3.4 we have an upper bound on $\mathbb{E}[\|m^t - \nabla f(x^t)\|_2^2]$, and we use the property $\sqrt{a_1 + a_2 + \cdots + a_n} \leq \sqrt{a_1} + \sqrt{a_2} + \cdots + \sqrt{a_n}$ to bound $\mathbb{E}[\|m^t - \nabla f(x^t)\|_2]$:

$$
\mathbb{E}\left[f(x^{t+1})\right] - \mathbb{E}\left[f(x^t)\right] \leq -\gamma \mathbb{E}\left[\left\|\nabla f(x^t)\right\|_1\right] + 2\sqrt{d}\gamma \mathbb{E}\left[\left\|m^t - \nabla f(x^t)\right\|_2\right] + \frac{dL\gamma^2}{2}
$$

$$
= -\gamma \mathbb{E}\left[\left\|\nabla f(x^t)\right\|_1\right] + \mathcal{O}\left[\frac{d^2}{1-\beta} \cdot L\gamma^2 + \sqrt{1-\beta}d\gamma\sigma + d\gamma L\tau \right.
$$

$$
\left. + \frac{d\gamma\Delta}{\tau} + \sqrt{d}\gamma\left(1 - \frac{1-\beta^2}{2d}\right)^{t/2}\left\|\nabla f(x^0)\right\|_2\right] + \frac{dL\gamma^2}{2}.
$$

Consequently, after summing from $t = 0$ to $t = T$, we obtain:

$$
\gamma \sum_{t=0}^{T} \mathbb{E}\left[\left\|\nabla f(x^t)\right\|_1\right] = \mathcal{O}\left[f(x^0) - f(x^T) + T \cdot \left(\frac{d^2}{1-\beta} \cdot L\gamma^2 + \sqrt{1-\beta}d\gamma\sigma + d\gamma L\tau\right)\right.
$$

$$
\left. + T \cdot \frac{d\gamma\Delta}{\tau} + \sqrt{d}\gamma \sum_{t=0}^{T}\left(1 - \frac{1-\beta^2}{2d}\right)^{t/2}\left\|\nabla f(x^0)\right\|_2\right]. \tag{13}
$$

Now, we divide equation (13) by $\gamma T$ from both sides and obtain:

$$
\frac{1}{T}\sum_{t=0}^{T}\mathbb{E}\left[\left\|\nabla f(x^t)\right\|_1\right] = \mathcal{O}\left[\frac{\delta_0}{\gamma T} + \frac{d\left\|\nabla f(x^0)\right\|_2}{T\sqrt{1-\beta}} + \frac{d^2 L\gamma}{1-\beta} + \sqrt{1-\beta}d\sigma + dL\tau + \frac{d\Delta}{\tau}\right],
$$

where we used a notation $\delta_0 := f(x^0) - f^*$. This finishes the proof. $\qquad\square$

# E    PROOFS FOR ZO MUON WITH JAGUAR (ALGORITHM 2)

## E.1    TECHNICAL LEMMAS

**Lemma E.1.** *Consider two arbitrary matrixes $A, B$ of the same shape and their SVD decomposition:* $A = U_A \Sigma_A V_A^T$, $B = U_B \Sigma_B V_B^T$. *Define $r_A$ and $r_B$ as ranks of $A$ and $B$, then it holds that*

$$
\left|\left\langle A, U_A V_A^T - U_B V_B^T\right\rangle\right| \leq 2\left\|A - B\right\|_{\mathcal{S}_1} \leq 2\sqrt{\operatorname{rank}(A - B)}\left\|A - B\right\|_F.
$$

*Proof.* We first provide an axillary notation:

$$
\delta := \left\langle A, U_A V_A^T - U_B V_B^T\right\rangle.
$$

Because $U_A$ and $V_A$ have orthonormal columns:

$$
\langle A, U_A V_A^\top\rangle = \operatorname{tr}\left(V_A \Sigma_A U_A^\top U_A V_A^\top\right) = \operatorname{tr}(\Sigma_A) = \|A\|_{\mathcal{S}_1}.
$$

Hence

$$
\delta = \|A\|_{\mathcal{S}_1} - \langle A, U_B V_B^\top\rangle.
$$

Insert $B$ and regroup:

$$
\delta = \|A\|_{\mathcal{S}_1} - \left(\langle B, U_B V_B^\top\rangle + \langle A - B, U_B V_B^\top\rangle\right) = \|A\|_{\mathcal{S}_1} - \|B\|_{\mathcal{S}_1} - \langle A - B, U_B V_B^\top\rangle.
$$

The first difference is controlled by the triangle inequality for the nuclear norm:

$$
\left|\|A\|_{\mathcal{S}_1} - \|B\|_{\mathcal{S}_1}\right| \leq \|A - B\|_{\mathcal{S}_1}.
$$

For the second term, Hölder's inequality with $\|U_B V_B^\top\|_2 = 1$ gives

$$
\left|\langle A - B, U_B V_B^\top\rangle\right| \leq \|A - B\|_{\mathcal{S}_1}.
$$

Therefore

$$
|\delta| \leq \|A - B\|_{\mathcal{S}_1} + \|A - B\|_{\mathcal{S}_1} = 2\|A - B\|_{\mathcal{S}_1}.
$$

Using the connection between the Frobenius ($\mathcal{S}_2$) by nuclear ($\mathcal{S}_1$) norms we obtain that:

$$|\delta| = \left\langle A, U_A V_A^T - U_B V_B^T \right\rangle \leq 2 \left\| A - B \right\|_{\mathcal{S}_1} \leq 2\sqrt{\operatorname{rank}(A - B)} \left\| A - B \right\|_F.$$

The factor 2 in the nuclear norm bound is sharp, as equality holds for $B = -A$. This finishes the proof. $\qquad\square$

We now provide lemma similar to the step Lemma 1 from (Sun et al., 2023), but in the matrix case.

**Lemma E.2** (Step lemma for Muon with momentum). *Let $f$ be an $L$-smooth function (Assumption 3.1), and let $X^\dagger, M \in \mathbb{R}^{m \times n}$ with $m \geq n$ be an arbitrary matrixes. We define*

$$X^\ddagger := X^\dagger - \gamma \cdot U_M V_M^T,$$

*where $\gamma > 0$ and $U_M V_M^T$ comes from SVD decomposition of $M$: $M = U_M \Sigma_M V_M^T$. Then, it holds that:*

$$f\left(X^\ddagger\right) - f\left(X^\dagger\right) \leq -\gamma \left\| \nabla f\left(X^\dagger\right) \right\|_{\mathcal{S}_1} + 2\sqrt{n}\gamma \left\| \nabla f\left(X^\dagger\right) - M \right\|_F + \frac{Ln\gamma^2}{2}.$$

*Proof.* The $L$-smoothness of the gradient (Assumption 3.1) gives us

$$f\left(X^\ddagger\right) - f\left(X^\dagger\right) \leq \left\langle \nabla f\left(X^\dagger\right), X^\ddagger - X^\dagger \right\rangle + \frac{L}{2} \left\| X^\ddagger - X^\dagger \right\|_F^2$$

$$= -\gamma \left\langle \nabla f\left(X^\dagger\right), U_M V_M^T \right\rangle + \frac{Ln\gamma^2}{2}$$

$$= -\gamma \left\langle \nabla f\left(X^\dagger\right), U_\nabla V_\nabla^T \right\rangle + \gamma \left\langle \nabla f\left(X^\dagger\right), U_\nabla V_\nabla^T - U_M V_M^T \right\rangle + \frac{Ln\gamma^2}{2},$$

where $U_\nabla V_\nabla^T$ comes from SVD decomposition of $\nabla f\left(X^\dagger\right)$: $\nabla f\left(X^\dagger\right) = U_\nabla \Sigma_\nabla V_\nabla^T$. Therefore the first dot product takes form:

$$-\gamma \left\langle \nabla f\left(X^\dagger\right), U_\nabla V_\nabla^T \right\rangle = -\gamma \operatorname{tr}\left(V_\nabla \Sigma_\nabla U_\nabla^T U_\nabla V_\nabla^T\right) = -\gamma \operatorname{tr}\left(\Sigma_\nabla\right) = -\gamma \left\| \nabla f\left(X^\dagger\right) \right\|_{\mathcal{S}_1}.$$

Now we utilize Lemma E.1 with $A = \nabla f\left(X^\dagger\right)$ and $B = M$:

$$f\left(X^\ddagger\right) - f\left(X^\dagger\right) \leq -\gamma \left\| \nabla f\left(X^\dagger\right) \right\|_{\mathcal{S}_1} + 2\gamma \left\| \nabla f\left(X^\dagger\right) - M \right\|_{\mathcal{S}_1} + \frac{Ln\gamma^2}{2}$$

$$\leq -\gamma \left\| \nabla f\left(X^\dagger\right) \right\|_{\mathcal{S}_1} + 2\sqrt{n}\gamma \left\| \nabla f\left(X^\dagger\right) - M \right\|_F + \frac{Ln\gamma^2}{2}.$$

This finishes the proof. $\qquad\square$

### E.2 PROOF OF THEOREM 3.7

*Proof.* We start from using Lemma E.2. For the points $X^t$, generated by Algorithm 2 it holds that:

$$f\left(X^{t+1}\right) - f\left(X^t\right) \leq -\gamma \left\| \nabla f\left(X^t\right) \right\|_{\mathcal{S}_1} + 2\sqrt{n}\gamma \left\| \nabla f\left(X^t\right) - M^t \right\|_F + \frac{Ln\gamma^2}{2}. \quad (14)$$

Now we take mathematical expectation of the both sides if (14) and bound the term $\mathbb{E}[\|\nabla f\left(X^t\right) - M^t\|_F]$ we again use Lemma 3.4 with $x^t = \overline{\operatorname{vec}}(X^t)$ and $m^t = \overline{\operatorname{vec}}(M^t)$. The result of Lemma 3.4 holds true with $d = m \cdot n$, since $\|A\|_F = \|\overline{\operatorname{vec}}(A)\|_2$. Therefore (14) takes form:

$$\mathbb{E}\left[f(X^{t+1})\right] - \mathbb{E}\left[f(X^t)\right] \leq -\gamma \mathbb{E}\left[\left\| \nabla f(X^t) \right\|_{\mathcal{S}_1}\right] + 2\sqrt{n}\gamma \mathbb{E}\left[\left\| M^t - \nabla f(X^t) \right\|_2\right] + \frac{nL\gamma^2}{2}$$

$$= -\gamma \mathbb{E}\left[\left\| \nabla f(X^t) \right\|_{\mathcal{S}_1}\right] + n^{1/2}\mathcal{O}\left[\frac{(mn)^{3/2}}{1 - \beta} \cdot L\gamma^2\right.$$

$$\left. + \sqrt{1 - \beta}(mn)^{1/2}\gamma\sigma + (mn)^{1/2}\gamma L\tau + \frac{(mn)^{1/2}\gamma\Delta}{\tau}\right.$$

$$+ n^{1/2}\gamma \left(1 - \frac{1-\beta}{mn}\right)^{t/2} \left\|\nabla f(X^0)\right\|_2 \Bigg] + \frac{nL\gamma^2}{2}.$$

$$= -\gamma\mathbb{E}\left[\left\|\nabla f(X^t)\right\|_{\mathcal{S}_1}\right] + \mathcal{O}\Bigg[\frac{m^{3/2}n^2}{1-\beta} \cdot L\gamma^2$$

$$+ \sqrt{1-\beta}m^{1/2}n\gamma\sigma + m^{1/2}n\gamma L\tau + \frac{m^{1/2}n\gamma\Delta}{\tau}$$

$$+ n^{1/2}\gamma \left(1 - \frac{1-\beta}{mn}\right)^{t/2} \left\|\nabla f(X^0)\right\|_2 \Bigg].$$

Consequently, after summing all $T$ steps, we obtain:

$$\gamma\sum_{t=0}^{T}\mathbb{E}\left[\left\|\nabla f(X^t)\right\|_{\mathcal{S}_1}\right] = \mathcal{O}\Bigg[f(X^0) - f(X^T)$$

$$+ T \cdot \left(\frac{m^{3/2}n^2}{1-\beta} \cdot L\gamma^2 + \sqrt{1-\beta}m^{1/2}n\gamma\sigma\right) \tag{15}$$

$$+ T \cdot \left(m^{1/2}n\gamma L\tau + \frac{m^{1/2}n\gamma\Delta}{\tau}\right)$$

$$+ n^{1/2}\gamma\sum_{t=0}^{T}\left(1 - \frac{1-\beta}{mn}\right)^{t/2}\left\|\nabla f(X^0)\right\|_2\Bigg].$$

Now, we divide equation (15) by $\gamma T$ from both sides and obtain:

$$\frac{1}{T}\sum_{t=0}^{T}\mathbb{E}\left[\left\|\nabla f(X^t)\right\|_{\mathcal{S}_1}\right] = \mathcal{O}\Bigg[\frac{\delta_0}{\gamma T} + \frac{m^{1/2}n\left\|\nabla f(x^0)\right\|_2}{T\sqrt{1-\beta}} + \frac{m^{3/2}n^2\gamma}{1-\beta} + \sqrt{1-\beta}m^{1/2}n\sigma$$

$$+ m^{1/2}nL\tau + \frac{m^{1/2}n\Delta}{\tau}\Bigg],$$

where we used a notation $\delta_0 := f(x^0) - f^*$. This finishes the proof. $\qquad\square$

## F  THE USE OF LARGE LANGUAGE MODELS (LLMs)

In this work, a LLM was used as a general-purpose assistant to help with drafting text and writing code. Specifically, the LLM assisted in generating initial code snippets, suggesting improvements to code structure, and formulating explanations in natural language. All content produced with the support of the LLM was carefully reviewed, edited, and validated by the authors to ensure correctness and originality. The authors take full responsibility for all the content presented in this paper.

