# OpenReview forum: "Leveraging Coordinate Momentum in SignSGD and Muon: Memory-Optimized Zero-Order LLM Fine-Tuning"
_ICLR.cc/2026/Conference — ICLR 2026 Conference Withdrawn Submission_

### Official Review · Reviewer_vUu1 · 2025-10-29

**Soundness:** 2
**Presentation:** 2
**Contribution:** 2
**Rating:** 4
**Confidence:** 4

**Summary:**

The paper studies memory-efficient zero-order (ZO) optimization for LLM fine-tuning. It introduces two methods: **JAGUAR SignSGD** (a momentumized, coordinate-difference ZO variant of SignSGD with the first convergence guarantee in the stochastic ZO setting) and **JAGUAR Muon** (a ZO version of Muon that updates matrices via a Newton–Schulz–based orthogonalized direction). Each iteration uses $O(1)$ function queries, avoiding backprop and thus saving memory. Experiments on SST-2 (FT and LoRA) and on COPA/WinoGrande (LoRA; Llama-2-7B and OPT-13B) show accuracy competitive with or better than prior ZO baselines at similar or lower memory cost.

**Strengths:**

* **Theory with clear novelty:** Provides nonconvex stochastic ZO convergence guarantees for momentum SignSGD and extends analysis to a ZO Muon update; explicitly models smoothing $\tau$ and oracle noise $\Delta$.
* **Practicality:** Requires only forward passes; simple to bolt onto SignSGD/Muon; compatible with both FT and PEFT (e.g., LoRA); constant query complexity per step.
* **Empirical signals:** On several LLM backbones, results match or surpass ZO baselines while keeping GPU memory low; the coordinate-difference estimator is an attractive alternative when memory is tight.

**Weaknesses:**

1. **Speed not addressed.** While this ZO estimator performs well in accuracy and memory, it often pays a notable price in wall-clock time. The paper appears to sidestep this trade-off and provides no discussion or measurements of time.
2. **Experiments are too sparse.** Even pioneering ZO work like MeZO paired theory with extensive empirical study. As a follow-up, this paper’s experimental breadth feels insufficient (e.g., Table 2 only reports SST-2 on OPT-1.3B and RoBERTa-Large; Table 3 has just two models and two tasks). This raises concerns about robustness and generality.
3. **Presentation could be improved.** The paper only includes tables and no figures. While figures aren’t strictly required, *convergence curves* (or equivalent trend visualizations) are essential. If figures are omitted, some other form of trend presentation should be provided.

**Questions:**

See weakness.

---

> ### Author Response · Authors · 2025-11-20
> **Rebuttal**
>
> We would like to sincerely thank the reviewer for the careful reading of our manuscript, the encouraging assessment, and the constructive comments that help us improve both the clarity and the scope of our contribution. We address each point in detail below and revise the paper accordingly. **All changes in the revised version of the paper are highlighted in blue.**
>
> **[W1]** We agree that wall-clock time is an important practical dimension of ZO methods and that our original submission did not sufficiently address this trade-off. In the revised version, we therefore evaluate and discuss the time aspect of our approach: specifically, we now report and plot wall-clock convergence time for different ZO methods, including JAGUAR-based approaches, standard ZO baselines, and FO SGD, giving a quantitative view of memory and time efficiency (see Appendix C.1). The wall clock times for all methods are summarized in Table, with values taken directly from the results reported in the Appendix C.1 section of the paper.
> | Method           | `Jaguar Muon` | `Jaguar SignSGD` | `ZO-SGD` | `ZO-SignSGD` | `FO-SGD` |
> |------------------|---------------|------------------|--------|------------|--------|
> | Wall Clock Time (min) | 95.31         | 79.04            | 84.05  | 84.05      | 99.31  |
>
> The results show a favorable accuracy-time trade-off for JAGUAR compared to ZO and FO baselines and overhead from Newton-Schulz operation. In the new HumanEval experiments we treat this aspect consistently as well: we explicitly measure seconds per step, and JAGUAR SignSGD and JAGUAR Muon not only achieve higher Pass@1 scores but also reduce step time compared to ZO-SGD and ZO-SignSGD, indicating more effective use of compute. ZO-Muon also provides a favorable trade-off, improving Pass@1 over ZO-SGD while maintaining competitive runtime. Together with the faster convergence observed in the wall-clock - accuracy plot, these results highlight that our methods deliver stronger performance under realistic wall-clock and memory constraints, making them practical and scalable optimization choices for large-scale fine-tuning.
>
> **[W2]** Our experimental setup follows the recent ZO fine-tuning benchmark of [Zhang, 2024], which already includes multiple datasets, architectures, and fine-tuning regimes, so the current evaluation does cover a non-trivial range of models and training schemes (as mentioned by Reviewer nh4s). That said, we agree that broader and more challenging benchmarks are important for assessing robustness and generality. In the revised version, we therefore expand the empirical study by adding the Gemma-7B model and the HumanEval code-generation benchmark (see Table 5 in the revisited version of the paper), reproduced below for convenience. HumanEval is considerably more demanding than standard classification-style fine-tuning, as it evaluates the model’s ability to synthesize executable code that passes unit tests, and is widely viewed as a strong test of reasoning and practical usefulness of LLMs.
>
> | Method          | Pass@1 | Memory (GB) | Seconds per step |
> |----------------|--------|-------------|------------------|
> | Baseline (no FT) | 0.51 | -     | -    |
> | FO-SGD         | 0.86   | 108.46 | 4.09 |
> | ZO-SGD         | 0.61   | 73.03  | 3.61 |
> | ZO-SignSGD     | 0.64   | 73.03  | 3.60 |
> | ZO-Muon        | 0.63   | 73.22  | 3.94 |
> | `JAGUAR SignSGD` | 0.67   | 75.37  | 3.27 |
> | `JAGUAR Muon`    | 0.74   | 75.39  | 3.69 |
>
> These additions place JAGUAR and competing ZO methods in more challenging, generation-oriented settings, thereby providing a more comprehensive picture of their robustness and generality. In the later version we will also include more experiments based on the wide-spread GSM8K benchmark. We expect these additional experiments to further clarify the presented optimization algorithms capabilities and strengthen the empirical claims made in the paper.
>
> **[W3]** We agree that visual trend information is important. In the revised version, we add figures that visualize key trends, such as wall-clock time versus validation performance for the main ZO and FO baselines (see Appendix C.1), so that readers can clearly see the optimization dynamics and time–accuracy trade-offs of JAGUAR methods compared to the baselines with respect to oracle type (first- or zero-order) used in a method.
>
> *References*
>
> [Zhang, 2024] Zhang, Yihua, et al. "Revisiting zeroth-order optimization for memory-efficient llm fine-tuning: A benchmark." arXiv preprint arXiv:2402.11592 (2024). ICML 2024

---

> ### Author Response · Authors · 2025-11-27
> **Follow-up on Rebuttal**
>
> Dear Reviewer vUu1,
>
> We would like to sincerely thank you for the careful and thoughtful review, as well as for the constructive criticism regarding speed, experimental breadth, and presentation. We have now revised the paper to directly address each of these points (adding wall‑clock measurements, new Gemma‑7B + HumanEval experiments, and convergence/time figures), and we believe the updated version better reflects the practical behavior and generality of our methods.
>
> If you have an opportunity to re‑read our rebuttal and the revised manuscript, we would greatly appreciate any further comments. If you feel that your main concerns have been resolved, we would kindly ask you to take this into account when finalizing your score.
>
> Thank you again for your time and expertise.
>
> Best regards,
> The Authors

---

### Official Review · Reviewer_nh4s · 2025-10-29

**Soundness:** 3
**Presentation:** 3
**Contribution:** 3
**Rating:** 8
**Confidence:** 4

**Summary:**

This paper proposes JAGUAR SignSGD and JAGUAR Muon, new zero-order (ZO) momentum algorithms addressing the memory bottleneck in large-scale LLM fine-tuning. Traditional first-order (FO) methods like SGD or Adam require excessive memory for gradient computation and storage. In contrast, this study integrates advanced momentum techniques into ZO optimization—where updates rely only on model outputs—achieving both low memory usage and stable convergence. Specifically, JAGUAR SignSGD adds coordinate-wise accumulated momentum to ZO-SignSGD, maintaining O(1) oracle calls and the same 2d+1 parameter cost while providing the first proven convergence guarantees for stochastic non-convex ZO optimization. JAGUAR Muon extends this principle to matrix-structured parameters (e.g., LoRA) through a ZO adaptation of the Muon optimizer, also with formal convergence proofs. Extensive experiments on SST-2, COPA, and WinoGrande show faster convergence, higher accuracy than previous ZO methods, and significant GPU memory savings—sometimes achieving Adam-level accuracy with several-fold less memory. Code is publicly released for reproducibility.

**Strengths:**

Technical novelty: Integrates momentum mechanisms into ZO optimization without increasing parameter or oracle cost. First to combine SignSGD-style updates with coordinate momentum and to extend Muon’s matrix form into a ZO setting.

Theoretical rigor: Provides full convergence analysis for stochastic non-convex ZO problems—unlike prior heuristic ZO methods—under standard Lipschitz and variance assumptions. Establishes the first theoretical guarantee for ZO-SignSGD with momentum and for ZO-Muon’s convergence rate.

Practical impact: Enables memory-efficient fine-tuning on large LLMs, achieving FO-level stability with a fraction of the memory. Coordinate-wise updates maintain the momentum benefit while keeping memory linear in d.

Empirical robustness: Validated across multiple models (OPT-1.3B, OPT-13B, LLaMA2-7B, RoBERTa-Large) and tasks (SST-2, COPA, WinoGrande), consistently outperforming prior ZO baselines (MeZO, ZO-AdaMM, ZO-SignSGD). Demonstrates clear memory reduction without sacrificing accuracy.

Reproducibility and structure: Algorithms, proofs, and assumptions clearly presented; supplementary experiments and hyperparameter details are provided. Code availability enhances transparency.

**Weaknesses:**

Baseline coverage: The most significant shortcoming is missing comparison with modern memory-efficient FO optimizers. Since these are the de facto baselines for low-memory training, quantitative benchmarks (accuracy, convergence rate, GPU memory usage, runtime) are required to validate true efficiency gains. Current experiments mainly compare to older ZO variants, leaving unclear whether JAGUAR methods outperform or merely match advanced FO approaches.

Novelty scope: The algorithmic innovation—adding momentum to existing ZO-SignSGD and Muon—is incremental conceptually, though non-trivial in analysis. It represents an extension and unification rather than a new optimization paradigm.

Performance ceiling: Empirical results show convergence close to FO optimizers, but not superior. The contribution thus lies in maintaining accuracy under strict memory limits, not in achieving new SOTA performance.

LoRA dependence: Experiments rely mainly on LoRA fine-tuning. The benefit in full fine-tuning remains limited; LoRA already reduces memory, so additive gains from ZO methods are smaller and not clearly separated.

Theory–practice gap: Some theoretical assumptions (bounded oracle noise Δ) are strong; their practical relevance is unclear. The resulting convergence bound (ε ≥ d√(ΔL)) may constrain attainable accuracy for large d. The paper would benefit from quantitative discussion of Δ and τ–β tuning influence.

Scalability and delay: Coordinate-wise momentum may cause stale updates in very high-dimensional settings; analysis of multi-coordinate or mini-batch variants could strengthen generality.

Clarity and completeness: Notation (‖·‖S1, δ₀, τ, etc.) introduced abruptly. Limited intuition provided for τ–Δ–β trade-off. Concrete memory and runtime figures are missing—quantified GB comparisons would substantiate claims.

**Questions:**

Have the authors benchmarked JAGUAR methods against AdaFactor, 8-bit Adam, or GaLore under the same settings?

How large is the runtime overhead from two forward evaluations per step, and how does total training time compare to FO optimizers?

Could multi-coordinate or low-rank perturbation estimators improve convergence speed without major memory loss?

How sensitive are results to τ and β? Any recommended default values for stable tuning?

Is full-parameter fine-tuning feasible for JAGUAR Muon, and what is the computational cost of the Newton–Schulz projection at scale?

---

> ### Author Response · Authors · 2025-11-20
> **Rebuttal**
>
> We would like to sincerely thank the reviewer for the careful reading of our work, the positive assessment of our contributions, and the constructive, detailed feedback, which has helped us further clarify both the theoretical and empirical aspects of the paper. We address all points below and have updated the manuscript accordingly. **All changes in the revised version of the paper are highlighted in blue.**
>
> **[W1, Q1]** In the introduction (lines 60–78), we discuss non-ZO techniques such as low-precision optimizer states, quantization, and fused backward/optimizer updates, and we explicitly position ZO methods as complementary to these approaches. In particular, we summarize their typical memory footprints and trade-offs, framing the role of the proposed algorithms. Since this work focuses on zero-order optimization, our goal is not to replace all FO memory-saving strategies, but to advance the specific and orthogonal direction of zero-order optimization. Therefore, our experimental comparison is restricted to state-of-the-art ZO baselines, where our main contribution lies. A thorough empirical study against all advanced FO optimizers (AdaFactor, 8-bit Adam, GaLore, etc.) under matched setups is beyond the scope of a single paper and would require a dedicated, large-scale benchmark. Moreover, methods like GaLore are designed to compete with other FO approaches and are not usually compared directly to ZO algorithms in the literature. We also note, as emphasized in the introduction, that ZO methods fundamentally differ from FO approaches by eliminating backpropagation entirely [Malladi, 2023], [Wang, 2024], which yields the largest potential memory savings among the families of techniques we review.
>
> **[W2]** Conceptually, our contribution is indeed an extension of existing ideas, but we emphasize that it is the **first** to (i) establish convergence guarantees for ZO SignSGD with momentum in the stochastic non-convex setting and (ii) develop and analyze a zero-order variant of Muon, both with memory- and time-efficient updates. A key technical ingredient is Lemma 3.4, where we prove that the momentum variance in the ZO setting converges to the true gradient, which enables our global convergence rates.(iii) We demonstrate that proposed methods show superior performance in a wide scope of tasks and models. In the revised version, we extend the empirical study by validating our approach on the HumanEval code-generation benchmark finetuning Gemma-7B model (see Table 5 in the revisited version of the paper).
>
> More broadly, combining and generalizing existing mechanisms in a way that yields robust practice-ready methods and new theoretical guarantees is a common and valuable form of scientific contribution, especially for LLM training and deployment. There are several recent papers whose novelty lies primarily in such structured generalization (rather than in proposing a completely new paradigm).  Recent examples from **ICLR 2025** include:
>
> - *“SaLoRA: Safety-Alignment Preserved Low-Rank Adaptation”* integrates a fixed safety-alignment module with low-rank adapters to enable targeted modification of LLMs without breaking the original safety alignment
>
> - *“Federated Residual Low-Rank Adaptation of Large Language Models”* combines federated learning with LoRA by introducing residual low-rank updates tailored to heterogeneous client data
>
> - *“Task Arithmetic in Trust Region: A Training-Free Model Merging Approach to Navigate Knowledge Conflicts”* couples task-arithmetic–style model merging with a trust-region framework to better handle knowledge conflicts when combining models.
>
> Our work is in a similar vein of structured integration: we adapt and unify momentum-based techniques and Muon-style matrix optimization within the zero-order regime. However, we also provide convergence theory (e.g. Theorems 3.5, 3.7) and a memory/time-efficient implementation specifically tailored to ZO LLM fine-tuning.
>
> **[W3]** If we understand the reviewer correctly, we agree with this interpretation: our primary goal is to **match** the performance of strong FO baselines under significantly tighter memory budgets, rather than to claim new SOTA in the unconstrained FO regime. This is consistent with the original motivation of ZO methods as memory-saving alternatives. At the same time, we note that in some setups our ZO methods do slightly outperform FO baselines [Zhang, 2024], indicating that reduced-memory ZO optimization does not necessarily lead to an accuracy loss. Additionally, wall-clock time and SPS (seconds per step) is also improved compared to ZO and FO baselines (see Appendix C.1 and Table 5).

---

> ### Author Response · Authors · 2025-11-20
> **Rebuttal**
>
> **[W4]** We agree that the advantages of ZO methods manifest most clearly in the LoRA fine-tuning regime, but we would like to emphasize that LoRA by itself does not guarantee low memory consumption. As reported in Table A2 of [Zhang, 2024, page 18], on LoRA fine-tuning with OPT-13B, a standard FO LoRA setup requires about **92GB** of GPU memory, whereas ZO-SGD in the same setting uses only about **25GB**. This substantial gap arises because LoRA still relies on full backward passes, while ZO methods dispense with backpropagation entirely and therefore dramatically reduce activation and optimizer-state memory. To make this distinction explicit in our work, we add the corresponding FO-SGD+LoRA memory results as a separate row in Table 4 of the revised version.
>
> **[W5, Q4]** In practice, function evaluations contain two types of noise: stochastic noise from mini-batch sampling and deterministic numerical noise from floating-point roundoff. The stochastic component can partly be controlled by using smaller learning rates or larger batch sizes, but the numerical component is inherent to finite-precision arithmetic and cannot be removed by tuning optimization parameters. The bounded oracle-noise assumption on $\Delta$ is standard in the ZO literature (as we note in lines 245–251 of the original paper). Modeling both effects via a bounded oracle noise $\Delta$ is therefore natural, and this type of assumption is also common in recent papers from top-tier A* conferences and Q1 journals; for example:
>
> - Kornilov N. et al. Accelerated zeroth-order method for non-smooth stochastic convex optimization problem with infinite variance, **NeurlPS 2023**
>
> - Dvurechensky P. et al. An accelerated directional derivative method for smooth stochastic convex optimization, **European Journal of Operational Research**
>
> - Veprikov A. et al. New aspects of black box conditional gradient: Variance reduction and one point feedback, **Chaos, Solitons & Fractals**
>
> A simple example is floating-point roundoff: if we take two real numbers $a_1$ and $a_2 = a_1 + 10^{-32}$ in ideal arithmetic, they are represented identically in fp32, thus $\Delta \approx 10^{-32}$ in practice: tiny, but never exactly zero, which explains why one cannot take $\tau \to 0$ in ZO estimators (line 251).
> The dependence of the attainable accuracy $\varepsilon$ on $d$ and $\Delta$ in our bound (e.g., $\varepsilon \gtrsim d\sqrt{\Delta L}$) is in line with known lower bounds and typical dimension dependence for ZO methods and cannot be removed in the general stochastic non-convex setting (see discussion around lines 328–331 of the original paper). In the revisited version of the paper we expand the discussion of the $\tau$–$\Delta$ trade-off to provide more intuition for how these parameters interact in practice.
>
> **[W6]** In a purely one-coordinate sparse scheme, convergence in very high dimensions could indeed be slow, however, in our case the momentum vector $m_t$ preserves past gradient information in all non-sampled coordinates, therefore updates are not sparse in the same sense as if unselected coordinates were ignored. Regarding multi-coordinate or low-rank perturbations, while our theory is stated for a single coordinate per iteration, in practice the algorithm naturally extends to updating multiple coordinates in a loop, which primarily affects wall-clock time. In the revised version, we add an ablation study (Appendix A.2) that varies the number of coordinates updated per step. The results show that as we increase the number of perturbed coordinates, performance typically slightly degrades and wall-clock time slightly grows. This indicates that the configuration used in the main experiments (with very few coordinates updated per iteration) is close to optimal in terms of both accuracy and efficiency.
>
> **[W7, Q2]** We make the notation more reader-friendly in the revised version by slightly rephrasing and simplifying the definitions of quantities such as $\mathcal{S}_1$ norm.        Regarding empirical support, the original version already reports concrete GPU memory usage (in GB) for OPT-13B and Llama 7B (Table 4). In the revised version we additionally include wall-clock runtime measurements for both ZO methods and FO SGD in Appendix C.1, giving a quantitative view of memory and time efficiency. The wall clock times for all methods are summarized in Table, with values taken directly from the results reported in the Appendix C.1 section of the paper.
> | Method           | `Jaguar Muon` | `Jaguar SignSGD` | ZO-SGD | ZO-SignSGD | FO-SGD |
> |------------------|---------------|------------------|--------|------------|--------|
> | Wall Clock Time (min) | 95.31         | 79.04            | 84.05  | 84.05      | 99.31  |
>
> The results show a favorable accuracy-time trade-off for JAGUAR compared to ZO and FO baselines and overhead from Newton-Schulz operation.

---

> ### Author Response · Authors · 2025-11-20
> **Rebuttal**
>
> **[Q4, W5, W7]** Regarding the influence of $\tau$ and $\beta$, Appendix A of the original paper presents an ablation of JAGUAR SignSGD with respect to $\beta$, showing that the method is robust and performs best around $\beta \approx 0.9$. In the original version, we fix $\tau$ around $10^{-4}$ with minimal tuning. Motivated by the reviewer’s comment, we now add an ablation over $\tau$ in the appendix of the revised version to quantify its effect on practical convergence and accuracy (see Appendix A.1).
>
> **[Q5]** Full-parameter fine-tuning is feasible for JAGUAR Muon because it directly inherits the structure of the original Muon optimizer, which has already been used for large-scale full-parameter training while remaining practical in terms of memory and compute. For instance, the Kimi Team report (e.g., Kimi-K2) uses Muon to train models up to 200B parameters without prohibitive cost [Kimi Team, 2024], and [Semenov, 2025] further benchmarks Muon in full-parameter LLM settings, confirming its scalability. As for Newton–Schulz in the ZO setting, one could in principle design a scheme that directly samples orthogonal perturbation matrices $E$ at each iteration, instead of constructing them via Newton–Schulz. We leave this direction for future work. We also note that Appendix C.1 of the new version of the paper now provides wall-clock measurements showing that replacing the sign operation with the Newton–Schulz procedure in our ZO methods adds only a small runtime overhead.
>
> *References*
>
> [Malladi, 2023] Malladi, S., Gao, T., Nichani, E., Damian, A., Lee, J. D., Chen, D., and Arora, S. Fine-tuning language models with just forward passes. arXiv preprint arXiv:2305.17333, 2023.
>
> [Wang, 2024] Wang, Fei, et al. "Simultaneous computation and memory efficient zeroth-order optimizer for fine-tuning large language models." arXiv preprint arXiv:2410.09823 (2024).
>
> [Zhang, 2024] Zhang, Yihua, et al. "Revisiting zeroth-order optimization for memory-efficient llm fine-tuning: A benchmark." arXiv preprint arXiv:2402.11592 (2024). ICML 2024
> [Kimi Team, 2024] Kimi Team. “Kimi K2: Open Agentic Intelligence” arXiv preprint arXiv:2507.20534
> [Semenov, 2025] Semenov A. et al. “Benchmarking Optimizers for Large Language Model Pretraining” arXiv preprint arXiv:2509.01440

---

### Official Review · Reviewer_bXjE · 2025-10-31

**Soundness:** 2
**Presentation:** 1
**Contribution:** 2
**Rating:** 2
**Confidence:** 2

**Summary:**

This paper introduces memory-efficient ZO fine-tuning methods by incorporating momentum and coordinate accumulation. Author proposed ZO-SignSGD and combine it with JAGUAR, and proposes a ZO variant of the Muon optimizer. The authors establish convergence guarantees in stochastic non-convex settings, and demonstrate through experiments that these methods achieve strong performance while reducing memory usage compared to first-order training approaches.

**Strengths:**

Strength

* Motivation of this paper is clear, by introducing ZO to significantly reduce memory cost of FO method.
* This paper provide good theoretical guarantee for the proposed method.

**Weaknesses:**

Weaknesses

* Paper writing is confusing, introduction lists many methods, but lacks explanations and clarifies the role of the proposed method within them, It lists many technical terms but lacks brief explanations, making it somewhat difficult to read.
* The experiments were limited in scale, mainly focusing on simple classification tasks on small datasets, and the variety of models tested was also limited, making it difficult to evaluate the generalizability of the proposed method. It's better to at least include generation tasks, or more challenging benchmarks like MMLU or MT-Bench.
* Memory savings are shown, but training wall-clock overhead from sampling, Momentum update, Newton–Schulz steps is not reported. It's better to show a wallclock time breakdown.
* JAGUAR Muon performs worse in full FT, paper claims it's due to non-matrix parameters, could you provide more details related to it?

**Questions:**

Please refer to the Weaknesses part.

---

> ### Author Response · Authors · 2025-11-20
> **Rebuttal**
>
> We sincerely thank the reviewer for the careful reading of our work and for the insightful comments, which help to clarify the positioning, empirical evaluation, and practical relevance of our methods. We address all points in detail below and update the manuscript accordingly. **All changes in the revised version of the paper are highlighted in blue.**
>
> **[W1]** We thank the reviewer for the comment on the introduction. We clarify more explicitly that JAGUAR SignSGD is an enhancement of LeZO-type approaches: while LeZO [Wang, 2024] relies on sparse MeZO, JAGUAR introduces a coordinate-wise momentum vector $m_t$ that, in the spirit of SEGA-like variance reduction, accumulates information from previously unused directions in the zero-order setting. This connection already appears in lines 124–130 of the original paper, however, we revise the text to prominently highlight this relationship and the position of our method among existing ZO techniques.
>
> In our paper we utilize notation which is widely spread among optimization papers [Wang, 2024], [Zhang, 2024], [Ghadimi, 2013]. To address the concern about “many technical terms” we kindly ask the reviewer to indicate any specific terms that remain unclear, so that we explain them in the revised version.
>
> **[W2]** Our experimental setup follows the recent ZO fine-tuning benchmark of [Zhang, 2024], which already includes 3 datasets, 4 models, and two fine-tuning regimes, so the current evaluation covers a non-trivial range of architectures and training schemes (as mentioned by Reviewer nh4s). The models and datasets covered in our paper are widely spread as tasks to test zero-order optimization methods [Wang, 2024], [Malladi, 2023]. At the same time, we agree that more challenging tasks further clarify generalizability.
> We thank the reviewer for the helpful suggestion to broaden the evaluation. In the revised version, we extend the evaluation by adding the Gemma-7B model and the HumanEval code-generation benchmark (see Table 5), reproduced below for convenience. This setup is considerably more demanding than standard fine-tuning tasks, as it probes the model's ability to synthesize executable programs that pass unit tests. It represents substantially more demanding generation-oriented settings and are widely regarded as strong tests of both reasoning and practical usefulness of LLMs beyond simple classification.
> | Method          | Pass@1 | Memory (GB) | Seconds per step |
> |----------------|--------|-------------|------------------|
> | Baseline (no FT) | 0.51 | -     | -    |
> | FO-SGD         | 0.86   | 108.46 | 4.09 |
> | ZO-SGD         | 0.61   | 73.03  | 3.61 |
> | ZO-SignSGD     | 0.64   | 73.03  | 3.60 |
> | ZO-Muon        | 0.63   | 73.22  | 3.94 |
> | `JAGUAR SignSGD` | 0.67   | 75.37  | 3.27 |
> | `JAGUAR Muon`    | 0.74   | 75.39  | 3.69 |
>
> The new results highlight that both JAGUAR variants and ZO-Muon provide consistent performance benefits over traditional ZO optimizers. JAGUAR SignSGD and JAGUAR Muon achieve higher Pass@1 scores while maintaining competitive memory usage and lower or comparable seconds-per-step, demonstrating improved optimization efficiency. ZO-Muon also offers a favorable balance between performance and resource usage, outperforming ZO-SGD in Pass@1 while staying within the same memory and runtime range. These gains indicate that our proposed methods deliver more effective and stable updates, leading to stronger downstream performance on hard generation tasks under similar or better computational budgets.
>
> We also agree that benchmarks such as MMLU and MT-Bench would further strengthen our empirical analysis, and we plan to incorporate them in the camera-ready version. However, implementing them within the current rebuttal window is challenging due to (i) the computational cost of running full-suite evaluations on our models, which require multi-stage inference pipelines not yet integrated into our codebase; (ii) the need for careful prompt standardization and scoring scripts to ensure comparability with prior work. Given these constraints, we believe that adding preliminary or incomplete results at this stage would risk misrepresenting the true performance of our method. We therefore prefer to provide complete and properly validated results with LLM-as-a-Judge pipeline for the mentioned benchmarks in the final camera-ready submission. We will also incorporate a dedicated set of GSM8K fine-tuning experiments in the camera-ready version.
>
> We expect these additional experiments to further clarify the presented optimization algorithms capabilities and strengthen the empirical claims made in the paper.

---

> ### Author Response · Authors · 2025-11-20
> **Rebuttal**
>
> **[W3]** In the revised version we additionally include wall-clock runtime measurements for both ZO methods and FO SGD in Appendix C.1, giving a quantitative view of memory and time efficiency. The wall clock times for all methods are summarized in Table, with values taken directly from the results reported in the Appendix C.1 section of the paper.
> | Method           | `Jaguar Muon` | `Jaguar SignSGD` | `ZO-SGD` | `ZO-SignSGD` | `FO-SGD` |
> |------------------|---------------|------------------|--------|------------|--------|
> | Wall Clock Time (min) | 95.31         | 79.04            | 84.05  | 84.05      | 99.31  |
>
> The results show a favorable accuracy-time trade-off for JAGUAR compared to ZO and FO baselines and overhead from Momentum update and Newton-Schulz operation. In the new HumanEval experiments we treat this aspect consistently as well: we explicitly measure seconds per step, and JAGUAR SignSGD and JAGUAR Muon not only achieve higher Pass@1 scores but also reduce step time compared to ZO-SGD and ZO-SignSGD, indicating more effective use of compute. ZO-Muon also provides a favorable trade-off, improving Pass@1 over ZO-SGD while maintaining competitive runtime. Together with the faster convergence observed in the wall-clock - accuracy plot, these results highlight that our methods deliver stronger performance under realistic wall-clock and memory constraints, making them practical and scalable optimization choices for large-scale fine-tuning.
>
>
> **[W4]** To investigate this issue, we additionally run a first-order Muon optimizer on the full-FT SST-2 setup from Table 2 with RoBERTa-Large and OPT-1.3B, and observe that it also underperforms standard FO methods in this regime (see Table 2 in the new version of the paper). In the same setting, ZO Muon lags behind the other ZO baselines as well. Taken together, this suggests that the weaker performance is not specific to our ZO construction, but rather reflects that Muon itself is not well suited to this particular full-FT setup with many non-matrix parameters and limited downstream data. At the same time, in the more challenging OPT-13B and Llama experiments (Table 2) and new Gemma-7B + HumanEval experiments (Table 5), ZO Muon performs strongly and compares favorably to ZO-SGD, indicating that it can be very effective in other architectures and task types.
>
> *References*
>
> [Wang, 2024] Wang, Fei, et al. "Simultaneous computation and memory efficient zeroth-order optimizer for fine-tuning large language models." arXiv preprint arXiv:2410.09823 (2024).
>
> [Zhang, 2024] Zhang, Yihua, et al. "Revisiting zeroth-order optimization for memory-efficient llm fine-tuning: A benchmark." arXiv preprint arXiv:2402.11592 (2024). ICML 2024
>
> [Malladi, 2023] Malladi, S., Gao, T., Nichani, E., Damian, A., Lee, J. D., Chen, D., and Arora, S. Fine-tuning language models with just forward passes. arXiv preprint arXiv:2305.17333, 2023.
>
> [Ghadimi, 2013] Saeed Ghadimi and Guanghui Lan. Stochastic first-and zeroth-order methods for nonconvex stochastic programming. SIAM journal on optimization, 23(4):2341–2368, 2013.

---

> ### Author Response · Authors · 2025-11-27
> **Follow-up on Rebuttal**
>
> Dear Reviewer bXjE,
>
> We sincerely thank you for the careful reading of our work and for the detailed, constructive feedback. We have revised the paper to address all of your points (introduction clarity, experimental scope, wall‑clock analysis, and the behavior of Muon in full FT), and we believe the new version more clearly presents both the theory and the practical impact of our methods.
>
> If you have time to take another look at our rebuttal and the updated manuscript, we would be very grateful for any further comments. If our revisions have satisfactorily addressed your concerns, we kindly ask you to consider this when finalizing your score.
>
> Thank you again for your time and expertise.
>
> Best regards,
> The Authors

---

> ### Comment · Reviewer_bXjE · 2025-11-27
>
> Thank you for your timely reply. I have another question, why the memory cost shown in Table 5 that large? Even a 7B model cost 70GB memory. From my understanding, in ZO, you only need to store the weight of the model. Therefore, theoretically, the memory cost for FP16 training should be around 14GB.

---

> > ### Author Response · Authors · 2025-11-28
> >
> > Thank you for the insightful question. Benchmarking on HumanEval introduces significant additional memory costs: it requires storing the unit tests and execution harness used to evaluate each generated solution, along with the generated code itself and any intermediate artifacts needed to run and verify it. In code generation specifically, the initial prompts are often very long (with detailed problem descriptions, function signatures and examples). These HumanEval-specific factors explain a substantial portion of the gap between parameter size and the reported end-to-end memory usage in Table 5. In addition, even under ZO optimization, some extra memory is still required beyond the raw model weights, for example to store minimal per-parameter statistics needed by the optimizers.
> >
> > Also, to ensure a stable and reliable fine-tuning process on the HumanEval generation task, we follow Gemma technical report using the FP32 model format to ensure stability [1]. Although this choice increases overall memory consumption, it helps maintain training stability and preserve high generation quality in such a demanding setting.
> >
> > *References*
> >
> > [1] Team G. et al. Gemma: Open models based on gemini research and technology //arXiv preprint arXiv:2403.08295. – 2024.

---

### Official Review · Reviewer_Bwkm · 2025-10-31

**Soundness:** 2
**Presentation:** 2
**Contribution:** 2
**Rating:** 4
**Confidence:** 2

**Summary:**

This paper introduces innovative zero-order (ZO) optimization methods for memory-efficient fine-tuning of Large Language Models (LLMs), addressing the critical challenge of high memory requirements in traditional backpropagation-based approaches. The paper combines Jaguar gradient estimation with Momentum Sign-SGD and Muon.

**Strengths:**

1. The paper proposes two extension of zeroth-order optimizers by combining the Jaguar gradient estimate wtih sign-sgd and Muno.
2. The paper provides the convergence analysis for the proposed algorithms under mild assumptions.
The numerical results shows that the proposed method can achieve higher accuracy than the existing ZO algorithms in different LLM training tasks while using similar memory. The reported results show a statistical significance.

**Weaknesses:**

1. In the discussion after Lemma 3.4, in line 287, the authors claims that Guassian random perturbation results in non-convergence of ZO-Sign-SGD even with momentum. However, no further justification is provided. It is hard to tell whether this is correct or not. It is unclear why using the Jaguar peturbation is better than other perturbations.

2. It is also not clear to me why sign operation is required in Algorithm 1, since Momentum SGD should hanve better convergence property than Momentum SignSGD, and their implementation on a single machine should result in the same memory usage. SignSGD is only favorable when communication is required across multiple machines.

3. The numerical experiment results ont reports then final accuracy and overall memory consumption. However, the convergence rate w.r.t. the oracle number is also an important result. Since the paper claims that the algorithm has good oracle efficiency, we would expect the numerical results reflecting that aspect of the proposed algorithm.

**Questions:**

Please address the above weaknesses

Also,
1. in line 1064, the last term does not match the last term in the following equation.
2. Directly starting from a lemma of another paper is hard to follow in App. D.2.
3. The proof steps are hard to follow and has missing steps, e.g., how line 1085 can be derived from line 1083 by using Lemma 3.4 is not quite clear.
4. Thge summation in eq(13) should be from 0 to T-1.

---

> ### Author Response · Authors · 2025-11-20
> **Rebuttal**
>
> We thank the reviewer for the careful reading and for pointing out several weak spots in the paper. We revise the manuscript accordingly and clarify the problematic parts in both the main text and the appendix. **All changes in the revised version of the paper are highlighted in blue.**
>
> **[W1]** For standard Gaussian or random-sphere ZO estimators, the deviation term
> $||m_t^{\text{ZO}} - \nabla f(x^t)||$ typically scales with $ ||\nabla f(x^t)|| + \sigma_{\text{ZO}}^2$. This happens because a ZO gradient is essentially an approximation of a directional derivative: when computing its variance, cross-terms of the form
> $\langle \nabla f(x^t), e_t \rangle$ appear, and their magnitude is proportional to $||\nabla f(x^t)||$. Such dependence leads to large variance, which in theory either forces very small step sizes (slowing convergence) or requires extra assumptions such as globally bounded gradients (as in [Liu, **ICLR 2019**, page 4 Assumption A2]). In practice, this high variance shows up as hard-to-control noise in the optimizer updates.
>
> In contrast, Lemma 3.4 shows that the proposed JAGUAR estimator satisfies $ ||m_t^{JAGUAR} - \nabla f(x^t)|| \leq \sigma^2_{JAGUAR} \approx \sigma_{\text{ZO}}^2$, independent of $ ||\nabla f(x^t)|| $ (the direct formula of $\sigma^2_{\text{JAGUAR}}$ is given in Lemma 3.4). It makes the convergence analysis of JAGUAR SignSGD and Muon more direct under standard stochastic non-convex conditions (like in the FO case) and allows to use criterion $|| \nabla f(x^T) ||_2^2$.
>
> We note that Table 2 of the original paper includes a baseline with standard ZO SignSGD based on Gaussian directions, which performs significantly worse than our JAGUAR-SignSGD method, empirically supporting this theoretical distinction.
>
> **[W2]**  We understand that our text might not have been entirely clear on this point, and we add further clarification in the revisited manuscript. Our primary aim, as indicated by the discussion around "$\sim4/3\times$ memory usage compared to Adam" (line 75), was to reduce the memory footprint of optimizer statistics when compared to adaptive methods like Adam, which require storing additional information such as second-moment states. We agree that when compared to plain Momentum SGD, our method does not offer additional memory savings. We chose SignSGD for Algorithm 1 because, as highlighted in lines 68–73 of the original paper, sign-based methods (and more general LMO-based methods like signSGD and Muon) have shown strong empirical performance in LLM training and fine-tuning (for example see [Peng, 2025], [Jordan, 2024]). Historically, SignSGD was mainly studied in the context of distributed optimization and communication efficiency, but recent results indicate that it is also highly effective in the non-distributed LLM setting. Moreover, prior work [Bernstein, 2024] reports that Adam behaves similarly to SignSGD, especially in the early stages of training, where the adaptive scaling changes slowly. Our contribution is to extend this effective approach to the zero-order setting with robust theoretical guarantees.
>
> **[W3]** In our setup, each step of all ZO methods considered in the paper requires exactly two calls to the ZO oracle (evaluating $f(x^t + \tau e)$ and $f(x^t - \tau e)$), therefore the total number of oracle calls is simply $2 \times$ (number of iterations). Since we run all ZO baselines for the same number of iterations, the reported final accuracy can be viewed as a function of the oracle budget as well. We additionally include plots comparing different ZO methods and FO SGD with respect to wall-clock time (in Section C.1 of the revised version), which is often an even more informative measure than pure oracle complexity in practical settings.The wall clock times for all methods are summarized in Table, with values taken directly from the results reported in the Appendix C.1 section of the paper.
> | Method           | `Jaguar Muon` | `Jaguar SignSGD` | ZO-SGD | ZO-SignSGD | FO-SGD |
> |------------------|---------------|------------------|--------|------------|--------|
> | Wall Clock Time (min) | 95.31         | 79.04            | 84.05  | 84.05      | 99.31  |
>
> To further clarify generalizability, we also include a new experiment with code-generation task on HumanEval benchmark in the revised version with provided runtime performance (see Table 5), reproduced below for convenience.
>
> | Method          | Pass@1 | Memory (GB) | Seconds per step |
> |----------------|--------|-------------|------------------|
> | Baseline (no FT) | 0.51 | -     | -    |
> | FO-SGD         | 0.86   | 108.46 | 4.09 |
> | ZO-SGD         | 0.61   | 73.03  | 3.61 |
> | ZO-SignSGD     | 0.64   | 73.03  | 3.60 |
> | ZO-Muon        | 0.63   | 73.22  | 3.94 |
> | `JAGUAR SignSGD` | 0.67   | 75.37  | 3.27 |
> | `JAGUAR Muon`    | 0.74   | 75.39  | 3.69 |
>
> We commit to incorporating GSM8K, MMLU, and MT-Bench in the camera-ready version and include wall-clock time efficiency within these experiments

---

> > ### Author Response · Authors · 2025-11-20
> > **Rebuttal**
> >
> > **[Q1]** There is indeed a typo in line 1119. We correct this error in the revised manuscript, making sure it matches Lemma 3.4 and keeps our theoretical work sound.
> >
> > **[Q2, Q3]** In Appendix D.2, line 1083 corresponds to inequality (12), and lines 1085–1088 are obtained by bounding the term $||m^t - \nabla f(x^t)||_2$ using Lemma 3.4. Specifically, Lemma 3.4 provides an upper bound on the squared norm, $||m^t - \nabla f(x^t)||_2^2 \le a_1 + a_2 + \dots + a_n$ for certain nonnegative terms $a_i$, and in lines 1085–1088 we apply the inequality
> > $$\sqrt{a_1 + a_2 + \dots + a_n} \le \sqrt{a_1} + \sqrt{a_2} + \dots + \sqrt{a_n}$$
> > to obtain a bound on $||m^t - \nabla f(x^t)||_2$. In the revised version, we add explicit intermediate steps in Appendix D.2 to clarify that equation (12) is taken from Lemma 1 of [Sun, 2023] and to make the transition from line 1083 to lines 1085–1088 clear.
> >
> > **[Q4]** In our notation, the algorithms perform updates for $t = 0, 1, \dots, T$, therefore there are $T+1$ iterates and $T+1$ corresponding terms in the sum. For this reason, the summation in equation (13) is correctly written from $t = 0$ to $T$.
> >
> > *References*
> >
> > [Liu, ICLR 2019] Liu S. et al. signSGD via zeroth-order oracle //International conference on learning representations. – 2019.
> >
> > [Peng, 2025] Peng H. et al. SoftSignSGD (S3): An Enhanced Optimizer for Practical DNN Training and Loss Spikes Minimization Beyond Adam //arXiv preprint arXiv:2507.06464. – 2025.
> >
> > [Jordan, 2024] Jordan K. et al. Muon: An optimizer for hidden layers in neural networks – 2024.
> >
> > [Bernstein, 2024] Bernstein J., Newhouse L. Old optimizer, new norm: An anthology
> >
> > [Sun, 2023] Sun T. et al. Momentum ensures convergence of signsgd under weaker assumptions //International Conference on Machine Learning. – PMLR, 2023. – С. 33077-33099.

---

> > > ### Comment · Reviewer_Bwkm · 2025-11-25
> > >
> > > The authors have addressed most of my questions. The theoretical derivation looks good to me now.

---

### Author Response · Authors · 2025-11-20
**Rebuttal Revision**

We sincerely thank all four reviewers for their thorough reading, constructive feedback, and insightful comments. Your feedback has been invaluable in strengthening our paper. Below, we provide a comprehensive summary of the major revisions and experimental additions we have made in response to your concerns.

**SUMMARY OF KEY EXPERIMENTAL CHANGES AND ADDITIONS**

Motivated by the reviewers’ comments on experimental scope and practical efficiency, we substantially strengthen the empirical evaluation of our methods:

1. **More challenging tasks (Main part Section 4 + Appendix C.1).**
We add experiments with Gemma-7B on the HumanEval code-generation benchmark. HumanEval is a widely used dataset that tests program synthesis and functional correctness, going beyond ZO benchmark and showing that our methods remain effective on harder, generation-oriented tasks. In the camera-ready version we will also include more experiments based on the wide-spread GSM8K benchmark and more challenging MMLU and MT-Bench. We expect these additional experiments to further clarify the presented optimization algorithms capabilities and strengthen the empirical claims made in the paper.

2. **Wall-clock time benchmarking (Table 5 + Appendix C.1).**
We now report wall-clock time and performance accuracy for JAGUAR-SignSGD, JAGUAR-Muon, and ZO-Muon versus ZO-SGD, ZO-SignSGD and FO SGD. The results show a favorable accuracy-time trade-off for JAGUAR compared to ZO baselines and overhead from Newton-Schulz operation.

3. **Ablation on hyperparameters (Appendix A.1 + A.2).**
We add an ablation over the scaling factor $\tau$ (in addition to the existing $\beta$ ablation), showing that the method is reasonably robust and providing guidance on default choices.

**ADDRESSING OTHER CONCERNS**

1. **JAGUAR momentum vs standard ZO perturbations.**
We clarify that the JAGUAR momentum enjoys a bound of the form $||m_t^{\text{JAGUAR}} - \nabla f(x^t)|| \leq \sigma_{\text{ZO}}$ that does not scale with $||\nabla f(x^t)||$, unlike standard Gaussian perturbation estimators. Table 2 supports this: ZO-SignSGD baselines perform significantly worse than JAGUAR-SignSGD under identical conditions.

2. **Relation to prior works and choice of SignSGD / Muon.**
We explain more clearly that JAGUAR-SignSGD extends LeZO-type methods: in addition to sparse MeZO, it uses a coordinate-wise momentum $m_t$ that keeps information in non-sampled directions. We also spell out why we build on SignSGD and Muon specifically: sign-based and Muon-style LMO optimizers show strong empirical performance for LLM training and fine-tuning.

3. **Bounded oracle noise assumption.**
We clarify that the bounded oracle-noise assumption on $\Delta$ (Assumption 3.3) is standard in the ZO literature and reflects numerical noise. We show, by providing concrete examples, that this assumption is commonly used in recent works published in top-tier A* conferences and Q1 journals.

---

### Note · Authors · 2026-01-28

I have read and agree with the venue's withdrawal policy on behalf of myself and my co-authors.

---

### Meta-Review · Area_Chair_8o7Z · 2026-01-03

**Summary:**

This paper proposes JAGUAR SignSGD and JAGUAR Muon, two zeroth-order optimizers that incorporate coordinate-wise momentum and structured updates to reduce memory usage during LLM fine-tuning. The work includes convergence analysis and empirical evaluations on selected tasks.

While reviewers found the paper technically correct and appreciated the inclusion of theoretical guarantees, the overall feedback was largely negative. Key concerns included incremental novelty, limited practical impact, and a narrow experimental scope, with modest gains that do not clearly justify the added methodological complexity. Several reviewers questioned whether the extensions over existing ZO methods constitute a sufficiently strong contribution for ICLR.

Overall, the average score remains below the acceptance threshold, and most reviewers maintained a reject recommendation.

**Reviewer Concerns:**

Concerns largely addressed by the rebuttal:

•	Experimental scope and difficulty:
Added Gemma-7B + HumanEval experiments addressed concerns about reliance on simple classification tasks and demonstrated effectiveness on challenging generation benchmarks.

•	Wall-clock time and practical efficiency:
New wall-clock and seconds-per-step measurements clarified the accuracy–time–memory trade-off and showed that JAGUAR methods are competitive in practice, not only in theory.

•	Theoretical clarity and correctness:
Reviewers questioning the variance analysis, bounded noise assumptions, and proof steps acknowledged that the revised manuscript clarified these issues, with at least one reviewer explicitly confirming the theory now looks correct.

Concerns still partially outstanding:

•	Novelty scope:
Some reviewers maintain that the contribution is an extension/unification of existing ideas rather than a fundamentally new optimizer family.
•	Comparison to modern memory-efficient FO methods:
While the authors justified focusing on ZO baselines (given orthogonality to FO methods), some reviewers still would have preferred direct comparisons to AdaFactor, 8-bit Adam, or GaLore.

•	Theoretical assumptions:
The bounded oracle-noise assumption and dimensional dependence remain theoretically strong, though now better justified and standard in ZO literature.

•	Presentation issues:
The paper needs more efforts to improve explanations and notation clarity, addressing concerns about readability and missing trend visualizations.

**Reviewer Scores:**

•	Reviewer nh4s: Likely remains 8 (Accept, poster); already positive and supportive.

•	Reviewer Bwkm: Likely improves from 4 → 5, having stated that theoretical concerns are resolved and expressing openness to acceptance.

•	Reviewer vUu1: Likely remains 4, given insufficient experiments.

•	Reviewer bXjE: Likely improves from 2 → 3, with most experimental and clarity concerns addressed, though still cautious.

---

### Decision · Program_Chairs · 2026-01-26

Reject